# $\mathtt{I}^2\mathtt{MoE}$: Interpretable Multimodal Interaction-aware Mixture-of-Experts

**Jiayi Xin**[1]  **Sukwon Yun**[2]  **Jie Peng**[3]  **Inyoung Choi**[1]  **Jenna L. Ballard**[1]
**Tianlong Chen**[2]  **Qi Long**[1]

## Abstract

Modality fusion is a cornerstone of multimodal learning, enabling information integration from diverse data sources. However, vanilla fusion methods are limited by **(1)** inability to account for heterogeneous interactions between modalities and **(2)** lack of interpretability in uncovering the multimodal interactions inherent in the data. To this end, we propose $\mathtt{I}^2\mathtt{MoE}$ (**I**nterpretable Multimodal **I**nteraction-aware **M**ixture **o**f **E**xperts), an end-to-end MoE framework designed to enhance modality fusion by explicitly modeling diverse multimodal interactions, as well as providing interpretation on a local and global level. First, $\mathtt{I}^2\mathtt{MoE}$ utilizes different interaction experts with weakly supervised interaction losses to learn multimodal interactions in a data-driven way. Second, $\mathtt{I}^2\mathtt{MoE}$ deploys a reweighting model that assigns importance scores for the output of each interaction expert, which offers sample-level and dataset-level interpretation. Extensive evaluation of medical and general multimodal datasets shows that $\mathtt{I}^2\mathtt{MoE}$ is flexible enough to be combined with different fusion techniques, consistently improves task performance, and provides interpretation across various real-world scenarios. Code is available at https://github.com/Raina-Xin/I2MoE.

## 1. Introduction

A core challenge in multimodal learning is modality fusion—the integration of information from multiple modalities to improve predictive performance (Baltrušaitis et al., 2019; Barnum et al., 2020; Lv et al., 2021). By leverag-

[1]University of Pennsylvania, PA, USA [2]University of North Carolina at Chapel Hill, NC, USA [3]University of Science and Technology of China, Anhui, China. Correspondence to: Qi Long <qlong@upenn.edu>, Tianlong Chen <tianlong@cs.unc.edu>, Jiayi Xin <jiayixin@seas.upenn.edu>.

*Proceedings of the 42ⁿᵈ International Conference on Machine Learning*, Vancouver, Canada. PMLR 267, 2025. Copyright 2025 by the author(s).

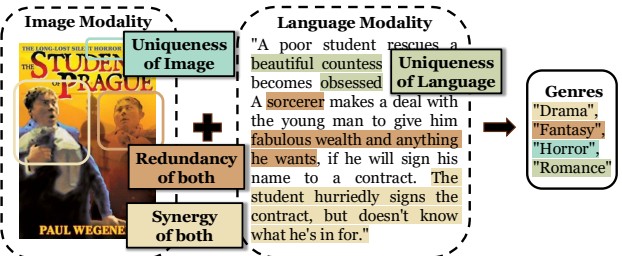

*Figure 1.* An illustrative example of modality interaction. The poster and plot are taken from the IMDB dataset.

ing diverse data sources such as text, images, audio, and sensor data, modality fusion enables the capture of intricate relationships across modalities, which is especially crucial in fields like healthcare, where accurate decision-making relies on multimodal insights (Liang et al., 2022b; Kline et al., 2022; Teoh et al., 2024). Although recent advancements in neural architectures, such as transformers (Vaswani et al., 2017; Tsai et al., 2019) and sparse mixture-of-experts (Shazeer et al., 2017; Fedus et al., 2022; Jin et al., 2024), have significantly improved the modeling of modality interactions, an important yet underexplored area is the systematic understanding of how modalities influence one another—whether they provide complementary, supplementary, or even conflicting information (Baltrušaitis et al., 2019; Liang et al., 2022b; 2023).

Understanding modality interaction is essential for advancing multimodal machine learning (Baltrušaitis et al., 2019; Liang et al., 2022b). An information-theoretic framework called Partial Information Decomposition (PID) (Wollstadt et al., 2023; Liang et al., 2023) offers a theoretical foundation for understanding modality interactions. PID decomposes information into four distinct types: *uniqueness for the first modality* (information specific to modality 1), *uniqueness for the second modality* (information specific to modality 2), *synergy* (emergent information arising from the combination of two modalities), and *redundancy* (shared information across two modalities).

Figure 1 illustrates the importance of carefully modeling different types of multimodal interactions. For instance, the **unique information** provided by the image modality

($\mathbf{m}_{img}$) contributes to predicting the Horror genre through distinct visual cues absent in the language modality ($\mathbf{m}_{lang}$) while the **unique information** from the language modality offers critical textual context for identifying the Romance genre. **Redundant information** refers to shared information present in both modalities, such as recognizing the Fantasy genre through the blurry figure in the poster and mentioning a "sorcerer" in the plot. Accurately classifying the movie as Drama, however, requires modeling **synergistic information** between the two modalities: visual elements such as clothing and facial expressions in $\mathbf{m}_{img}$, complement the narrative details from $\mathbf{m}_{lang}$. From this example, systematic modeling of multimodal interactions is needed to make accurate predictions.

While the PID framework provides valuable theoretical insights into the proportions of different modality interactions within a dataset, its practical application is limited, lacking integration into **end-to-end** and **interpretable** deep learning frameworks. Most existing multimodal fusion methods do not *explicitly* model multimodal interactions (Liu et al., 2018; Tsai et al., 2019; Xue & Marculescu, 2023). Notable efforts to address this gap, such as (Wörtwein et al., 2022; Yu et al., 2024; Dufumier et al., 2024), exhibit key limitations: they either focus exclusively on pairwise modality interactions (Wörtwein et al., 2022), require separate estimates for each interaction type (Yu et al., 2024), or lack sufficient interpretability (Dufumier et al., 2024). The opportunity to directly leverage PID for improving both task performance and model interpretability within multimodal fusion frameworks remains largely unexplored.

In contrast to earlier works, we propose $\mathtt{I}^2\mathtt{MoE}$, an end-to-end mixture-of-experts (MoE) framework designed to enhance task performance while improving interpretability. Our approach incorporates separate parameters and weakly-supervised interaction losses, enabling the mixture of interaction experts to effectively model diverse interactions between modalities. To further enhance interpretability, we introduce a re-weighting model that assigns importance scores to each interaction expert, providing insights into decision-making at both local (sample-level) and global (dataset-level) scales. $\mathtt{I}^2\mathtt{MoE}$ is backbone-agnostic and can be seamlessly integrated with any modality fusion approach. We evaluate the effectiveness of $\mathtt{I}^2\mathtt{MoE}$ on two medical datasets and three real-world multimodal datasets, demonstrating its ability to consistently improve performance while offering interpretable insights into the model's decision-making process for individual samples.

Our contributions are summarized as follows:

★ We introduce $\mathtt{I}^2\mathtt{MoE}$, a novel mixture-of-experts framework designed to explicitly model diverse modality interactions through specialized parameters and weakly-supervised interaction losses, enabling a more nuanced understanding of multimodal data.

★ We enhance interpretability by providing both sample-level and dataset-level insights into model decisions, offering a deeper understanding of how interaction experts contribute to predictions.

★ $\mathtt{I}^2\mathtt{MoE}$ is highly flexible and can be seamlessly integrated with existing modality fusion methods, demonstrating its versatility in improving vanilla multimodal fusion backbones.

★ Extensive experiments on five diverse real-world multimodal datasets validate the efficacy of $\mathtt{I}^2\mathtt{MoE}$, showcasing significant performance improvements (up to $5.5\%$ in accuracy) and interpretability benefits over vanilla modality fusion methods.

## 2. Related Work

**Modality Interaction** is theoretically grounded in the Partial Information Decomposition (PID) framework (Liang et al., 2023), which analyzes heterogeneous interactions but lacks an end-to-end learning framework. Prior works attempt to model interactions but are either restricted to specific interaction types (Zhang et al., 2023; Kim et al.), fail to quantify interactions in the data (Wörtwein et al., 2024; Liang et al., 2024; Long et al., 2024; Dufumier et al., 2024), or are limited to only two modalities (Wörtwein et al., 2022; Fan et al., 2024). Our approach bridges this gap by directly modeling and quantifying modality interactions within a unified MoE-based fusion architecture, enabling effective and interpretable multimodal learning.

**Multimodal Fusion** integrates data from multiple sources to enhance prediction tasks. Existing methods often rely on concatenating input modalities using off-the-shelf architectures (Liu et al., 2018; Tsai et al., 2019; Xue & Marculescu, 2023; Shazeer et al., 2017; Fedus et al., 2022). Mixture-of-Experts (MoE) offers a natural architecture for modeling interactions via expert specialization (Jacobs et al., 1991; Chen et al., 1999; Yuksel et al., 2012). Several recent works (Mustafa et al., 2022; Lin et al., 2024; Yu et al., 2024) explore MoE for multimodal learning. Among them, only MMoE (Yu et al., 2024) explicitly models different types of modality interactions by using a mixture of interaction experts on sentiment analysis. However, MMoE treats modality interaction modeling as a preprocessing step rather than integrating it into an end-to-end learning framework, limiting flexibility and interpretability.

**Multimodal Interpretation** has gained traction as researchers seek to explain decision-making in multimodal AI systems. Prior studies either focus on isolating the effect of individual modalities while overlooking inter-modal interactions (Ismail et al., 2022; Ghosh et al., 2023; Swamy et al., 2024b), provide human-interpretable rationales but

fail to quantify interaction contributions (Park et al., 2018; Zadeh et al., 2018; Dominici et al., 2023), or lack explicit categorization of interaction types (Tsai et al., 2020; Chefer et al., 2021; Lyu et al., 2022; Liang et al., 2022a; Wenderoth et al., 2024). As no prior work has explored interpretation from a modality interaction perspective, our contribution is to systematically quantify multimodal interactions while maintaining interpretability.

# 3. Interpretable Multimodal Interaction-aware Mixture-of-Experts

## 3.1. Preliminary and Notation

**Problem Setup.** Let $\mathcal{M} = \{\mathbf{m}_1, \mathbf{m}_2, \ldots, \mathbf{m}_n\}$ denote a set of $n$ input data modalities, and let $\mathbf{y}$ represent the target variable for a given task. For classification tasks, $\mathbf{y}$ is expressed as a one-hot encoded vector corresponding to the class label. For regression tasks, $\mathbf{y}$ is a real-valued scalar. The objective is twofold: **(1)** to improve the performance of predicting the ground truth target $\mathbf{y}$ by effectively modeling the interactions between modalities in $\mathcal{M}$, and **(2)** to provide meaningful interpretations of these multimodal interactions.

**Vanilla multimodal fusion** (Figure 2(a)) utilizes modality-specific encoders $\mathcal{E} = \{\mathrm{E}_1, \mathrm{E}_2, \ldots, \mathrm{E}_n\}$ to process $\mathcal{M}$ and obtain latent embeddings $\mathcal{L} = \{\mathbf{e}_1, \mathbf{e}_2, \ldots, \mathbf{e}_n\}$, where each embedding is computed as $\mathbf{e}_i = \mathrm{E}_i(\mathbf{m}_i)$ for $i \in \{1, \ldots, n\}$. We define the fusion method as $\mathrm{F}$, which operates on the set of latent embeddings $\mathcal{L}$ and produces a fused embedding $\mathbf{x}$, expressed as: $\mathrm{F}(\mathcal{L}) = \mathbf{x}$. A prediction head $\mathrm{H}$ maps the fused embedding to the final prediction, expressed as: $\mathrm{H}(\mathbf{x}) = \hat{y}$. However, this naive modality fusion approach does not explicitly account for the heterogeneous interactions present between $\mathcal{M}$.

## 3.2. Algorithm Overview of $\mathrm{I}^2\mathrm{MoE}$ Framework

$\mathrm{I}^2\mathrm{MoE}$ is a mixture of interaction experts, where each expert specializes in modeling a specific type of multimodal interaction. The predictions from individual interaction experts are weighted by a re-weighting model to produce the final prediction. **During the training phase**, we first perform a forward pass using the intact input of all modalities to estimate the multimodal prediction. Next, additional forward passes are conducted, where one modality is replaced by a random vector in each pass. These perturbed inputs serve as weak supervision signals to help train the interaction experts to specialize in different types of modality interactions. We designed a **dual-objective loss**, encouraging the interaction experts to specialize effectively without degrading task performance. The task loss is calculated using the re-weighted output from the interaction experts with the complete modality input, while the interaction loss is computed from the outputs generated with the perturbed

modality inputs. **During inference**, a single forward pass is performed using the complete modality input. The final output is a weighted sum of the interaction expert prediction with the weights produced by the re-weighting model (Equation 1). We provide a detailed explanation of $\mathrm{I}^2\mathrm{MoE}$ with two input modalities in Section 3.3, describe its extension to a higher number of modalities in Section 3.4, and explain how to obtain multimodal interaction interpretation in Section 3.5.

## 3.3. $\mathrm{I}^2\mathrm{MoE}$ with Two Input Modalities

### 3.3.1. $\mathrm{I}^2\mathrm{MoE}$ ARCHITECTURE

Figure 2(b) illustrates the $\mathrm{I}^2\mathrm{MoE}$ architecture for modeling different types of modality interactions in two input modalities. We employ a MoE comprising four fusion models, referred to as *interaction experts*: $\mathrm{F}_{\mathrm{uni1}}$, $\mathrm{F}_{\mathrm{uni2}}$, $\mathrm{F}_{\mathrm{syn}}$, and $\mathrm{F}_{\mathrm{red}}$. Each interaction expert specializes in capturing a specific type of interaction: $\mathrm{F}_{\mathrm{uni1}}$ models the unique information contained in modality $\mathbf{m}_1$; $\mathrm{F}_{\mathrm{uni2}}$ models the unique information contained in modality $\mathbf{m}_2$; $\mathrm{F}_{\mathrm{syn}}$ captures the synergistic information between $\mathbf{m}_1$ and $\mathbf{m}_2$; and $\mathrm{F}_{\mathrm{red}}$ models the redundant information between $\mathbf{m}_1$ and $\mathbf{m}_2$.

Each interaction expert processes the latent embeddings of the two modalities, $\mathbf{e}_1$ and $\mathbf{e}_2$, and produces fused embeddings, represented as $\mathbf{x}_i = \mathrm{F}_i(\mathbf{e}_1, \mathbf{e}_2)$, where $i \in \{\mathrm{uni1}, \mathrm{uni2}, \mathrm{syn}, \mathrm{red}\}$. These fused embeddings are then passed through a prediction head within each interaction expert, generating predictions for the corresponding interaction type as $\hat{\mathbf{y}}_i = \mathrm{H}_i(\mathbf{x}_i)$, where $i \in \{\mathrm{uni1}, \mathrm{uni2}, \mathrm{syn}, \mathrm{red}\}$. To combine the predictions from the four interaction experts, we introduce a re-weighting model $\mathrm{W}$, which assigns importance scores to the predictions of each expert. The model $\mathrm{W}$ takes the latent embeddings $\mathbf{e}_1$ and $\mathbf{e}_2$ as inputs and outputs a set of soft weights: $\mathrm{W}(\mathbf{e}_1, \mathbf{e}_2) = [w_{\mathrm{uni1}}, w_{\mathrm{uni2}}, w_{\mathrm{syn}}, w_{\mathrm{red}}]$. The final prediction is obtained by combining the predictions from all experts using these weights, expressed as:

$$\hat{\mathbf{y}} = \sum_i w_i \cdot \hat{\mathbf{y}}_i, \quad i \in \{\mathrm{uni1}, \mathrm{uni2}, \mathrm{syn}, \mathrm{red}\}. \quad (1)$$

### 3.3.2. $\mathrm{I}^2\mathrm{MoE}$ LEARNING OBJECTIVE

The loss function consists of two components. The first component is the *task loss*, which encourages the predictions of $\mathrm{I}^2\mathrm{MoE}$, $\hat{\mathbf{y}}$, to closely match the ground truth target $\mathbf{y}$. The second component, termed the *interaction loss*, ensures that the initially identical fusion models within $\mathrm{I}^2\mathrm{MoE}$ specialize into interaction experts by capturing diverse interactions in the dataset.

Following Yu et al. (2024), we characterize interaction types by comparing unimodal and multimodal predictions: predictions made using only the first modality ($\mathbf{y}_1$), predictions

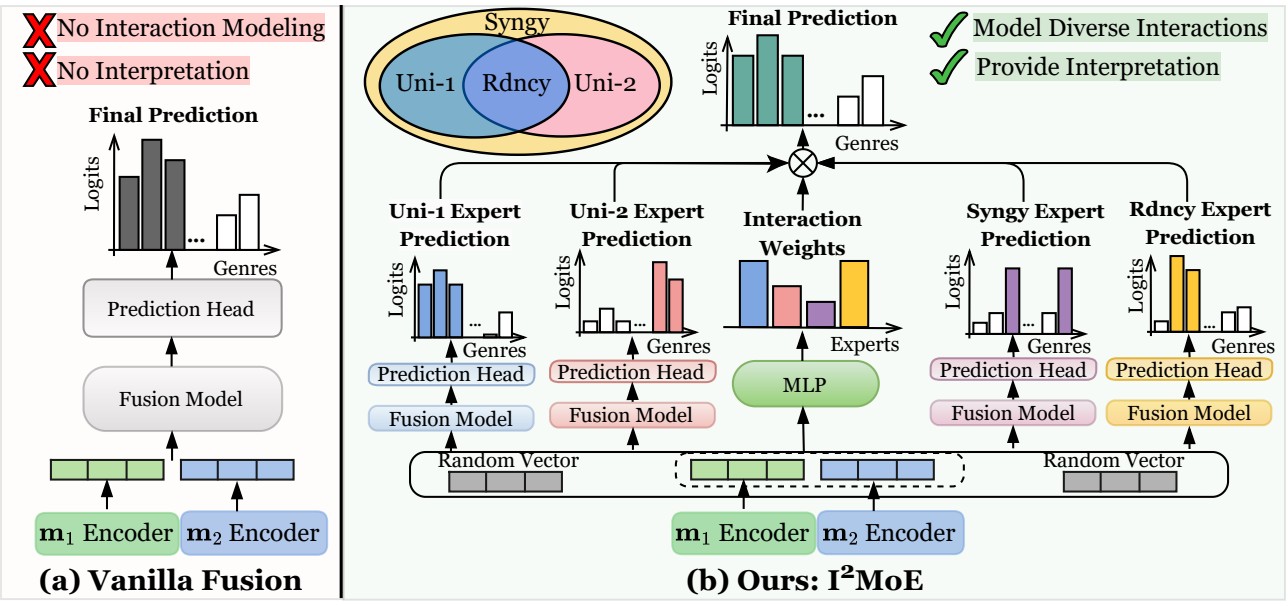

*Figure 2.* Comparison between vanilla modality fusion and I²MoE in the case of movie genre classification with two input modalities.
**Left**: Existing modality fusion approaches typically use the same parameters to model all types of interactions between the two modalities.
**Right**: In contrast, we design a mixture-of-experts framework that employs four different interaction experts and a re-weighting model to explicitly capture heterogeneous interactions between the two input modalities.

made using only the second modality ($\mathbf{y}_2$), and predictions made using both modalities ($\mathbf{y}_{12}$). For interactions emphasizing the uniqueness of the first modality, the relationships are defined as $\mathbf{y}_{12} = \mathbf{y}_1$ and $\mathbf{y}_{12} \neq \mathbf{y}_2$. Similarly, for interactions emphasizing the uniqueness of the second modality, we have $\mathbf{y}_{12} = \mathbf{y}_2$ and $\mathbf{y}_{12} \neq \mathbf{y}_1$. For synergistic interactions, the condition is $\mathbf{y}_{12} \neq \mathbf{y}_1$ and $\mathbf{y}_{12} \neq \mathbf{y}_2$. For redundant interactions, the relationship is $\mathbf{y}_{12} = \mathbf{y}_1 = \mathbf{y}_2$.

To approximate the interaction loss, we simulate the unimodal scenario by replacing one of the modalities with a random vector. For each interaction expert, a unimodal prediction using only the first modality can be obtained by replacing the latent embedding of the second modality with a random vector $\mathbf{r}$, represented as:

$$\hat{\mathbf{y}}_{-2,i} = \mathrm{H}_i\big(\mathrm{F}_i(\mathrm{E}_1(\mathbf{x}_1), \mathbf{r})\big), \qquad (2)$$

where $i \in \{\mathrm{uni1}, \mathrm{uni2}, \mathrm{syn}, \mathrm{red}\}$. Similarly, a unimodal prediction using only the second modality can be generated by replacing the latent embedding of the first modality with $\mathbf{r}$, expressed as:

$$\hat{\mathbf{y}}_{-1,i} = \mathrm{H}_i\big(\mathrm{F}_i(\mathbf{r}, \mathrm{E}_2(\mathbf{x}_2))\big), \qquad (3)$$

where $i \in \{\mathrm{uni1}, \mathrm{uni2}, \mathrm{syn}, \mathrm{red}\}$.

We designed a general framework to approximate different types of modality interactions. In all cases, the output using the complete multimodal input, $\hat{\mathbf{y}}_{12}$, serves as the anchor. For the $\mathrm{F}_{\mathrm{uni1}}$, the output with modality 2 masked, $\hat{\mathbf{y}}_{-2}$, is

treated as a positive example, while the output with modality 1 masked, $\hat{\mathbf{y}}_{-1}$, is treated as a negative example. The objective is to encourage $\hat{\mathbf{y}}_{12}$ to be maximally similar to $\hat{\mathbf{y}}_{-2}$ and maximally different from $\hat{\mathbf{y}}_{-1}$, since $\mathrm{F}_{\mathrm{uni1}}$ models the uniqueness information presented in $\mathbf{m}_1$. For the $\mathrm{F}_{\mathrm{uni2}}$, $\hat{\mathbf{y}}_{-2}$ is treated as a negative example, while $\hat{\mathbf{y}}_{-1}$ is treated as a positive example. Here, the objective is to encourage $\hat{\mathbf{y}}_{12}$ to be maximally similar to $\hat{\mathbf{y}}_{-1}$ and maximally different from $\hat{\mathbf{y}}_{-2}$, since $\mathrm{F}_{\mathrm{uni2}}$ models the uniqueness information presented in $\mathbf{m}_2$. For the $\mathrm{F}_{\mathrm{syn}}$, $\hat{\mathbf{y}}_{-1}$ and $\hat{\mathbf{y}}_{-2}$ are both treated as negative examples. The objective is to ensure that $\hat{\mathbf{y}}_{12}$ is maximally different from both $\hat{\mathbf{y}}_{-2}$ and $\hat{\mathbf{y}}_{-1}$, capturing interactions that require the combination of both modalities. For the $\mathrm{F}_{\mathrm{red}}$, $\hat{\mathbf{y}}_{-1}$, and $\hat{\mathbf{y}}_{-2}$ are treated as positive examples. The goal is to encourage $\hat{\mathbf{y}}_{12}$, $\hat{\mathbf{y}}_{-2}$, and $\hat{\mathbf{y}}_{-1}$ to be as similar as possible, modeling information shared between the modalities. We discuss the connection between the proposed interaction loss and the PID formulation in Appendix A and present empirical evidence supporting the design choice of random vector masking, in Appendix B.

### 3.4. Extend I²MoE to Higher Number of Modalities

**Increase Uniqueness Interaction Experts.** To extend I²MoE to support more than two input modalities, we increase the number of interaction experts to the $|\mathcal{M}| + 2$. Instead of a combinatorial explosion in the number of interaction experts, as the number of input modalities grows, we define $m$ uniqueness interaction experts, one for each input

---

**Algorithm 1** Training and Inference of $\mathtt{I^2MoE}$

---

**Require:** Modalities $X_1, \ldots, X_n$, label $T$
**Require:** Modality-specific Encoders $\{\text{Enc}_i\}_{i=1}^n$
**Require:** Experts $\{F_i\}_{i=1}^E$, reweighting module $W$
**Require:** Expert loss functions $\{\text{InteractionLoss}_i\}_{i=1}^E$
    *// Training with masked modality input*
1: Encode modalities: $Z_i \leftarrow \text{Enc}_i(X_i)$ for $i = 1, \ldots, n$
2: **for** $i = 1$ to $E$ **do**
3:    $[\hat{y}_i^{(0)}, \ldots, \hat{y}_i^{(n)}] \leftarrow F_i^{\text{multi}}(Z_1, \ldots, Z_n)$
4:    $L_{\text{int}}^i \leftarrow \text{InteractionLoss}_i(\hat{y}_i^{(0)}, \hat{y}_i^{(1:n)})$
5: **end for**
6: $[w_1, \ldots, w_E] \leftarrow W(Z_1, \ldots, Z_n)$
7: $\hat{y} \leftarrow \sum_{i=1}^E w_i \cdot \hat{y}_i^{(0)}$
8: $L_{\text{task}} \leftarrow \ell(\hat{y}, T)$
9: $L_{\text{total}} \leftarrow L_{\text{task}} + \frac{\lambda_{\text{int}}}{E} \sum_{i=1}^E L_{\text{int}}^i$
10: Update model parameters to minimize $L_{\text{total}}$
11: **procedure** INFERENCE
12:    Encode modalities: $Z_i \leftarrow \text{Enc}_i(X_i)$ for $i = 1, \ldots, n$
13:    $\hat{y}_i^{(0)} \leftarrow F_i(Z_1, \ldots, Z_n)$ for $i = 1, \ldots, E$
14:    $[w_1, \ldots, w_E] \leftarrow W(Z_1, \ldots, Z_n)$
15:    $\hat{y} \leftarrow \sum_{i=1}^E w_i \cdot \hat{y}_i^{(0)}$
16:    Store $\{\hat{y}_i\}$, $w$, and prediction $\hat{y}$
17: **end procedure**

---

modality, along with a single synergy expert and a single redundancy expert. Each uniqueness expert, $F_{\text{uni},i}$, is responsible for capturing the unique information specific to its corresponding modality, $m_i \in \mathcal{M}$, where $i \in \{1, \ldots, n\}$. The synergy expert, $F_{\text{syn}}$, focuses on modeling global synergistic interactions across all modalities, while the redundancy expert, $F_{\text{red}}$, captures globally redundant information shared among the modalities.

**Modify Interaction loss.** For uniqueness expert $i$, we consider the output of the complete modality as the anchor. The masked modality $i$ serves as a negative example, while all other perturbed inputs are treated as positive examples. This is because the unique information of modality $i$ is lost when the modality embedding is replaced by random vectors. For the synergy interaction loss, we treat all the output of the perturbed modality as negative examples, as input modality perturbations damage the synergistic information. For the redundancy interaction loss, we consider the output of the perturbed modality as a positive example because redundant information remains available even when one modality is masked. For classification tasks, we employed Triplet Margin Loss to model uniqueness interactions. For synergy and redundancy interactions, we utilized Cosine Similarity to capture the relationships between modality outputs. For regression tasks, we used the Mean Squared Error (MSE) Loss to measure differences in predictions.

$\mathtt{I^2MoE}$ **Algorithm and Complete Objective.** We present the training and inference pipeline of $\mathtt{I^2MoE}$ in Algorithm 1. The complete learning objective is provided in Appendix C. We analyze computational overhead and scalability in Appendix D.

### 3.5. Local and Global Interpretation from $\mathtt{I^2MoE}$

Local interpretation provides insight into the extent to which different interactions contribute to the final prediction for each individual sample, while global interpretation highlights the average trends of interaction importance across the entire dataset. For $\mathtt{I^2MoE}$, decisions are made locally for each specific input sample by analyzing the prediction, $\hat{y}_i$, from each interaction expert $F_i$, and the importance coefficients, $w_i$, assigned by the MLP-based re-weighting model W. Global interpretation for $\mathtt{I^2MoE}$ can be achieved by calculating the statistics of the importance weights $w_i$ assigned to each interaction expert across all samples in the test set, thereby capturing the overall trends in feature contributions.

## 4. Experiment Setup

**Data Collection and Datasets.** We evaluate our method on five multimodal datasets, using all available modalities while discarding samples with missing data. Two Medical Multimodal Datasets: ▷ **ADNI** (Weiner et al., 2010; 2017) consists of 2,380 samples for Alzheimer's Disease classification (Dementia, Cognitively Normal, or Mild Cognitive Impairment). It includes four modalities: Image ($\mathcal{I}$), Genetic ($\mathcal{G}$), Clinical ($\mathcal{C}$), and Biospecimen ($\mathcal{B}$). ▷ **MIMIC-IV** (Johnson et al., 2023) is a critical care dataset with 9,003 patient records for one-year mortality prediction (binary classification), utilizing three modalities: Lab ($\mathcal{L}$), Notes ($\mathcal{N}$), and Code ($\mathcal{C}$). Three General Multimodal Datasets: ▷ **IMDB** (Arevalo et al., 2017) includes 25,959 movies for multi-label genre classification across 23 genres, leveraging Image ($\mathcal{I}$) and Language ($\mathcal{L}$) modalities. ▷ **MOSI** (Zadeh et al., 2016) comprises 2,199 annotated YouTube clips for sentiment analysis (regression with scores $\in$ [-3,3] and then map to binary classification), incorporating Vision ($\mathcal{V}$), Audio ($\mathcal{A}$), and Text ($\mathcal{T}$) modalities. ▷ **ENRICO** (Leiva et al., 2020) contains 1,460 Android app screens for UI design classification into 20 categories, featuring two modalities: Screenshot ($\mathcal{S}$) and Wireframe ($\mathcal{W}$). Detailed dataset preprocessing is provided in Appendix E.

**Modality-Specific Encoders and Prediction Heads.** The primary objective of our experiments is to evaluate whether the proposed mixture-of-experts framework improves modality fusion. To ensure a fair comparison, we control for variations in modality-specific encoders (E) and prediction models (H) by using the same E and H for both vanilla multimodal fusion and $\mathtt{I^2MoE}$. For further details on the encoder and classification head configurations, please refer to Appendix F.

**Baseline Fusion Methods.** To validate the effectiveness of $\mathrm{I}^2\mathtt{MoE}$ in enhancing multimodal learning, we compare it to various widely used fusion techniques. We begin with fundamental approaches, including early fusion (EF) (Baltrušaitis et al., 2019), late fusion (LF) (Baltrušaitis et al., 2019), low-rank multimodal fusion (LRMF) (Liu et al., 2018), and multimodal transformers (MulT) (Tsai et al., 2019). We then implement more advanced fusion methods, including interpretable conditional computation (InterpretCC) (Swamy et al., 2024a), the Switch Transformer (SwitchGate) (Fedus et al., 2022), and sparse mixture-of-experts (MoE++) (Jin et al., 2024). In both SwitchGate and MoE++, the MLP layer in MulT is replaced with a sparse MoE layer that incorporates the respective routing function.

**Implementations.** The dataset is partitioned into training, validation, and testing sets, with 70% allocated for training, 15% for validation, and the remaining 15% for testing. Each experiment is run three times with different random seeds and the results are averaged. To ensure a fair comparison with other baselines, we utilize the optimal hyperparameter settings provided in the original studies. If a dataset does not have reported optimal parameters, we perform a grid search over the key hyperparameters of the baseline methods. The re-weighting model (W) is implemented as a multilayer perceptron (MLP). For a detailed description of the hyperparameter settings, we refer the reader to Appendix G.

# 5. Performance and Interpretability of $\mathrm{I}^2\mathtt{MoE}$

## 5.1. $\mathrm{I}^2\mathtt{MoE}$ Demonstrates Superior Task Performance

In Table 1, we compared the performance of $\mathrm{I}^2\mathtt{MoE}$ combining with MulT ($\mathrm{I}^2\mathtt{MoE-MulT}$) with other vanilla fusion methods across five datasets: ❶ Compared to vanilla MulT, $\mathrm{I}^2\mathtt{MoE}$ yields a significant accuracy improvement of **5.5%** for ADNI and **3%** for MOSI, demonstrating its ability to enhance the performance of existing transformers. ❷ Across all datasets, $\mathrm{I}^2\mathtt{MoE}$ outperforms advanced baselines such as SwitchGate and MoE++, with a notable gain of **2.5%** accuracy, **1.5%** AUROC on ADNI, and **1.4%** improvement in Macro F1 for IMDB. ▷ These results illustrate the benefit of $\mathrm{I}^2\mathtt{MoE}$ in tackling the challenges of modality interaction to achieve superior task performance.

## 5.2. Generalization Across Different Fusion Methods

To evaluate the generalizability of $\mathrm{I}^2\mathtt{MoE}$ across various fusion backbones, we integrate it with three fusion architectures, including MoE++, SwitchGate, and Interpret-CC, and assess the combined models on all datasets (Table 2): ❶ For the **ADNI dataset**, $\mathrm{I}^2\mathtt{MoE}$ yields significant performance gains, with up to **5.23%** improvement in accuracy and **2.12%** in AUROC when combined with SwitchGate. ❷

On the **MIMIC dataset**, $\mathrm{I}^2\mathtt{MoE}$ achieves notable AUROC improvements of **4.43%** when combined with Interpret-CC, highlighting its ability to capture complex interaction in multimodal patient data. However, accuracy decreases (-0.56% to -11.82%) are observed, which can be attributed to dataset imbalance. In such cases, the model becomes less overfitted to the majority class, leading to a decrease in accuracy but a corresponding increase in AUROC, reflecting improved performance in distinguishing between classes overall. ❸ $\mathrm{I}^2\mathtt{MoE}$ consistently enhances multimodal learning, achieving improvements in Micro F1 on **IMDB** (**2.45%**), sentiment analysis accuracy on **MOSI** (**4.76%**), and design classification accuracy on **ENRICO** (**5.14%**) when integrated with MoE++ and SwitchGate. ▷ Results with different fusion backbones emphasize the generalizability and effectiveness of $\mathrm{I}^2\mathtt{MoE}$.

## 5.3. $\mathrm{I}^2\mathtt{MoE}$ Offers Local Interpretation

To illustrate the interpretability provided by $\mathrm{I}^2\mathtt{MoE}$ on the individual sample level, we present a qualitative example from the IMDB test set where $\mathrm{I}^2\mathtt{MoE-MulT}$ makes a correct prediction (Figure 3). This example showcases how different interaction experts contribute to the final prediction through visualized logits and assigned weights, offering a clear decomposition of the decision-making process. The ground truth genres of this movie include `Animation`. In Figure 3(a), the logits produced by each interaction expert are shown. Notably, the uniqueness expert for the image modality and the redundancy expert generate positive logits, while the synergy expert yields a negative logit. This aligns with the visual content of the image, which features cartoon characters uniquely contributing to the prediction in Figure 3(d). Figure 3(b) depicts the weights assigned by the reweighting mechanism. Higher weights are given to the uniqueness expert for the image modality and the redundancy expert. As shown in Figure 3(c), the final weighted logits for the `Animation` genre become positive, enabling the correct prediction. This example demonstrates how $\mathrm{I}^2\mathtt{MoE}$ leverages different interaction patterns to make accurate predictions. We provide human evaluation of local interpretation in Appendix H and additional qualitative examples in Appendix I.

## 5.4. $\mathrm{I}^2\mathtt{MoE}$ Enables Global Interpretation

We analyze the weight assigned by the reweighting model to each interaction expert across all test samples. Figure 4 illustrates the weight variation across datasets, offering insights into dataset-level interaction patterns. The reweighting model demonstrates the ability to adaptively assign distinct weights to interaction experts, reflecting its capacity to capture dataset-specific nuances. In the **ADNI dataset**, weights are relatively uniform, with a subtle bias toward certain experts, indicating balanced contributions from all inter-

*Table 1.* Comparison of Accuracy, AUROC, and F1 scores across different fusion methods and datasets. The upper panel lists vanilla fusion methods, while the last row presents the proposed `I`$^2$`MoE` framework combined with MulT fusion method.

| Dataset | ADNI | | MIMIC | | IMDB | | MOSI | ENRICO |
|---|---|---|---|---|---|---|---|---|
| Metrics | **Accuracy** | **AUROC** | **Accuracy** | **AUROC** | **Micro F1** | **Macro F1** | **Accuracy** | **Accuracy** |
| EF | $52.01_{\pm 0.92}$ | $65.69_{\pm 1.81}$ | $67.63_{\pm 1.66}$ | $67.75_{\pm 0.93}$ | $56.10_{\pm 0.27}$ | $41.12_{\pm 1.08}$ | $72.16_{\pm 0.66}$ | $42.35_{\pm 0.81}$ |
| LF | $50.79_{\pm 3.11}$ | $68.60_{\pm 3.77}$ | $67.11_{\pm 1.06}$ | $67.58_{\pm 0.88}$ | $56.22_{\pm 0.03}$ | $45.27_{\pm 0.64}$ | $70.51_{\pm 1.14}$ | $44.20_{\pm 1.64}$ |
| LRMF | $50.79_{\pm 2.20}$ | $69.37_{\pm 3.13}$ | $70.17_{\pm 1.79}$ | $65.45_{\pm 6.31}$ | $56.22_{\pm 0.03}$ | $45.27_{\pm 0.64}$ | $\mathbf{76.63}_{\pm 0.18}$ | $46.12_{\pm 1.06}$ |
| InterpretCC | $54.53_{\pm 3.43}$ | $72.18_{\pm 1.70}$ | $72.34_{\pm 4.48}$ | $61.93_{\pm 2.53}$ | $58.00_{\pm 0.23}$ | $48.68_{\pm 0.11}$ | $75.85_{\pm 0.07}$ | $47.60_{\pm 1.56}$ |
| SwitchGate | $\underline{62.28}_{\pm 1.17}$ | $\underline{79.70}_{\pm 0.20}$ | $70.98_{\pm 0.83}$ | $68.26_{\pm 3.25}$ | $55.92_{\pm 0.07}$ | $47.33_{\pm 0.47}$ | $72.35_{\pm 0.27}$ | $43.95_{\pm 2.83}$ |
| MoE++ | $58.08_{\pm 2.52}$ | $75.18_{\pm 1.95}$ | $72.51_{\pm 2.09}$ | $68.50_{\pm 2.13}$ | $58.15_{\pm 0.32}$ | $50.49_{\pm 0.25}$ | $70.85_{\pm 0.83}$ | $\underline{47.83}_{\pm 1.86}$ |
| MulT | $59.57_{\pm 0.66}$ | $77.21_{\pm 0.51}$ | $\mathbf{72.42}_{\pm 2.53}$ | $\underline{68.79}_{\pm 3.34}$ | $\underline{59.68}_{\pm 0.19}$ | $\underline{51.41}_{\pm 0.04}$ | $68.80_{\pm 0.78}$ | $47.37_{\pm 1.82}$ |
| `I`$^2$`MoE-MulT` | $\mathbf{65.08}_{\pm 1.52}$ | $\mathbf{81.09}_{\pm 0.02}$ | $69.78_{\pm 0.91}$ | $\mathbf{68.81}_{\pm 0.99}$ | $\mathbf{61.00}_{\pm 0.44}$ | $\mathbf{52.38}_{\pm 0.48}$ | $71.91_{\pm 2.20}$ | $\mathbf{48.22}_{\pm 1.61}$ |

*Table 2.* Comparison of metrics across datasets using different fusion methods for `I`$^2$`MoE`. Performance improvements are indicated in blue, and decreases are indicated in red.

| Dataset | i$^2$MoE- | SwitchGate | InterpretCC | MoE++ |
|---|---|---|---|---|
| **ADNI** | **Accuracy** | 67.51 (5.23) | 56.02 (1.49) | 59.01 (0.93) |
| | **AUROC** | 81.82 (2.12) | 73.36 (1.18) | 75.69 (0.51) |
| **MIMIC** | **Accuracy** | 70.42 (-0.56) | 69.85 (-2.49) | 60.69 (-11.82) |
| | **AUROC** | 69.08 (0.82) | 66.36 (4.43) | 69.15 (0.65) |
| **IMDB** | **Micro F1** | 57.43 (1.51) | 58.32 (0.32) | 60.60 (2.45) |
| | **Macro F1** | 47.77 (0.44) | 49.21 (0.53) | 50.73 (0.24) |
| **MOSI** | **Accuracy** | 73.86 (1.51) | 76.14 (0.29) | 75.61 (4.76) |
| **ENRICO** | **Accuracy** | 49.09 (5.14) | 49.09 (1.49) | 47.83 (0) |

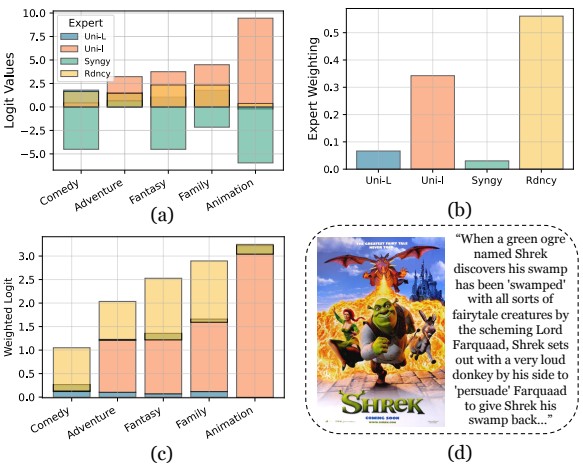

*Figure 3.* Qualitative example of local interpretation on the IMDB dataset provided by `I`$^2$`MoE-MulT`. Ground truth labels are `Comedy`, `Adventure`, `Fantasy`, `Family`, and `Animation`. (a) Logits output by different interaction experts. (b) Weighting assigned by the reweighting model. (c) Contribution of each interaction expert to the final weighted logit. (d) Raw image and language modalities used for prediction.

action experts to the model's performance. Conversely, the **MIMIC dataset** displays pronounced variability in weight assignments, emphasizing `I`$^2$`MoE` 's reliance on reweighting

model to address variance among individual patients. For the **IMDB dataset**, the weight variation is less pronounced compared to MIMIC, aligning with its more homogeneous characteristics. The **MOSI dataset** shows evenly distributed weights, reflecting equal contributions from all interaction experts. Finally, the **ENRICO dataset** demonstrates a concentrated weight distribution with dominant experts for the screenshot modality.

## 6. In-depth Analysis of `I`$^2$`MoE`

### 6.1. Accuracy of Individual Experts

To further analyze the effectiveness of `I`$^2$`MoE`, we compare its task performance against individual interaction experts across different datasets, as shown in Figure 5. The results highlight the following insights: ❶ Across all datasets, the overall performance of `I`$^2$`MoE-MulT` (red horizontal line) consistently surpasses that of any individual interaction expert expert, with performance gains of 2.2%, 1.3%, 7.1%, 0.6%, and 2.6% for **ADNI**, **MIMIC**, **IMDB**, **MOSI**, and **ENRICO**, respectively. ▷ This underscores the advantage of leveraging a mixture-of-experts approach over single-expert methods. ❷ The proposed method exhibits the largest performance gains in datasets with high interaction importance distribution variability, such as **MIMIC** and **ENRICO**. While for more uniform datasets like **MOSI**, the performance of individual experts is closer to that of the overall model, indicating that the ensemble effect may be less pronounced in these cases. ▷ This suggests that the fusion of multiple experts becomes particularly beneficial in datasets with complex and heterogeneous multimodal interactions.

### 6.2. Interaction Expert Diversification

To analyze the diversification of different interaction experts, we evaluate the ratio of expert agreement to disagreement and assess the corresponding accuracy of `I`$^2$`MoE`. A

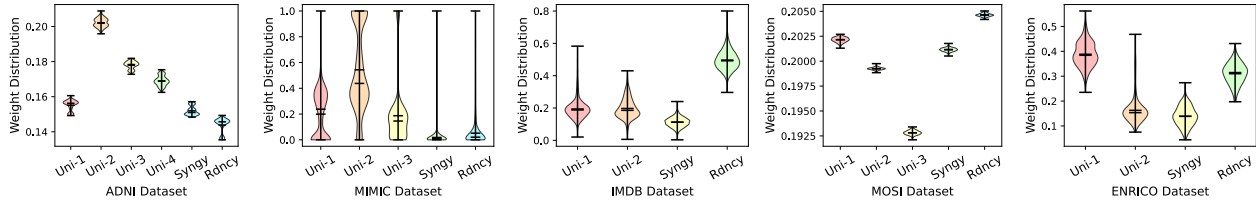

*Figure 4.* Visualization of interaction weight distributions across all test samples for five datasets. Black bars indicate the median, mean, and extreme values.

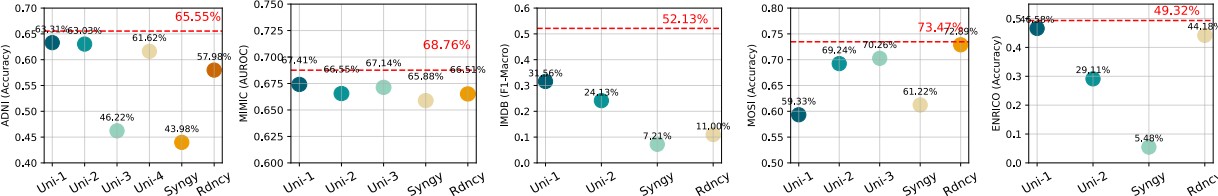

*Figure 5.* Comparison between the task performance of I$^2$MoE-MulT (red horizontal line) and each individual interaction expert across different datasets.

high proportion of disagreement among experts indicates greater diversity, which is essential for capturing distinct interaction patterns. Furthermore, when experts disagree, we expect I$^2$MoE to still maintain a high level of accuracy, demonstrating its ability to leverage diverse expert opinions effectively.

Table 3 presents the proportion of cases where experts disagree or agree, along with the corresponding accuracy of I$^2$MoE across five datasets: ❶ For **ADNI** and **MIMIC** datasets, the proportion of disagreement among experts is relatively high (**81%** and **85%**, respectively), while I$^2$MoE achieves correct predictions in a substantial portion of these cases. ❷ On the **IMDB** and **ENRICO** datasets, the proportion of disagreement is very high (**99.99%** and **98%**), yet I$^2$MoE achieves significantly fewer correct predictions when experts disagree (**15.85% Correct, 84.14% Wrong** and **46.85% Correct, 51.44% Wrong**). ❸ For the **MOSI** dataset, the disagreement proportions (**59%**) highlight moderate diversity among experts. Notably, I$^2$MoE maintains relatively high accuracy when experts disagree (**37.80% Correct for MOSI**. ▷ These results indicate a potential need for better handling of disagreement in complex datasets, and how dataset characteristics influence the diversification and effectiveness of interaction experts.

# 7. Ablation Studies

To validate the effectiveness of I$^2$MoE, we perform extensive ablation studies by systematically removing or modifying key components of the model. Each variant is designed to assess the contribution of specific design choices to the overall performance: **(1)** No-Interaction: The

*Table 3.* Interaction experts agreement analysis on test set for all datasets. "Disagree" or "Agree" indicates whether all expert prediction is the same. ✓ ("Correct") or ✗ ("Incorrect") refers to the correctness of I$^2$MoE's prediction.

| % of Data | ADNI | MIMIC | IMDB | MOSI | ENRICO |
|---|---|---|---|---|---|
| Disagree, ✓ | 48.74 | 63.51 | 15.85 | 37.80 | 46.85 |
| Disagree, ✗ | 32.40 | 21.39 | 84.14 | 21.97 | 51.44 |
| Agree, ✓ | 16.34 | 6.37 | 0.00 | 34.11 | 1.37 |
| Agree, ✗ | 2.52 | 8.73 | 0.01 | 6.12 | 0.34 |

interaction loss is removed, resulting in a simple mixture-of-experts model without explicit encouragement for learning diverse multimodal interaction among experts. **(2)** Latent-Contrastive: The interaction loss is applied directly to the latent embeddings produced by each interaction expert instead of their outputs. **(3)** Simple-Weight: The MLP-based reweighting model is replaced by a shared, learnable global weight that does not adapt to individual samples. **(4)** Less-Forward: Perturbation is reduced by randomly masking only two modalities per sample instead of perturbing all modalities. **(5)** Synergy-Redundancy: Only synergy and redundancy experts are included, omitting uniqueness experts.

From Table 7: ❶ **No-Interaction:** Removing the interaction loss results in significant performance degradation across all datasets (e.g., -6.35% accuracy on ADNI and -3.99% AUROC), confirming that explicitly encouraging diversity among experts is crucial for capturing complementary modality interactions. ❷ **Latent-Contrastive:** Applying the interaction loss to latent embeddings instead of expert outputs causes a noticeable performance drop (e.g., -6.91% accuracy on ADNI). This highlights the importance of applying the interaction loss at the output level to di-

*Table 4.* Ablation study results on three datasets (ADNI, MOSI, ENRICO), showing the impact of removing or modifying key components of $\text{I}^2\text{MoE}$. Each row corresponds to a variant of the model with a specific component ablated. Performance drops (in red) are reported relative to the full model.

| Dataset | ADNI | | MOSI | ENRICO |
|---|---|---|---|---|
| Ablation | Accuracy | AUROC | Accuracy | Accuracy |
| (1) | 58.73 (-6.35) | 77.10 (-3.99) | 69.49 (-2.42) | 47.63 (-0.59) |
| (2) | 58.17 (-6.91) | 75.40 (-5.69) | 69.68 (-2.23) | 47.50 (-0.72) |
| (3) | 59.29 (-5.79) | 74.55 (-6.54) | 68.46 (-3.45) | 47.49 (-0.73) |
| (4) | 59.76 (-5.32) | 76.81 (-4.28) | 69.89 (-2.02) | 46.92 (-1.30) |
| (5) | 56.77 (-8.31) | 74.30 (-6.79) | 70.12 (-1.79) | 47.49 (-0.73) |

rectly guide expert specialization. ❸ **Simple-Weight:** Replacing the sample-specific reweighting model with a global weight reduces performance (e.g., -5.32% accuracy on ADNI and -1.30% on ENRICO), demonstrating the value of adaptive reweighting for leveraging diverse expert outputs effectively. ❹ **Less-Forward:** Reducing modality perturbations leads to reduced accuracy (e.g., -5.79% on ADNI and -3.45% on MOSI). This suggests that generating sufficient negative examples through extensive perturbation is essential for capturing diverse interactions. ❺ **Synergy-Redundancy:** Limiting the experts to only synergy and redundancy results in the largest performance drop (e.g., -8.31% accuracy on ADNI). This emphasizes the importance of uniqueness experts in modeling comprehensive modality interactions. ▷ The ablation study demonstrates that each component of $\text{I}^2\text{MoE}$ is vital for its success.

## 8. Conclusion

We introduced $\text{I}^2\text{MoE}$, a novel MoE framework designed to enhance multimodal task performance and interpretability by explicitly capturing heterogeneous modality interactions. Extensive experiments on five real-world datasets demonstrated the superiority of $\text{I}^2\text{MoE}$ in improving performance across diverse multimodal scenarios. By leveraging a mixture-of-experts design with adaptive reweighting and specialized interaction losses, our approach systematically models and quantifies modality interactions. Additionally, we analyzed the distribution of interaction weights, providing meaningful insights at both the sample and dataset levels, which enhances the interpretability of the model's predictions. We also conducted ablation studies to evaluate the impact of each design component and demonstrated the flexibility of $\text{I}^2\text{MoE}$ to generalize across various fusion methods. For future work, alternative forms of interaction loss could be explored to further improve performance. Additionally, integrating feature attribution methods to analyze the contributions of individual features within interaction experts can offer deeper interpretable insights.

## Acknowledgements

This work was supported in part by NIH grants, RF1AG063481, R01AG071174, and U01CA274576. The content is solely the responsibility of the authors and does not necessarily represent the official views of the NIH. We would like to thank the anonymous reviewers for their insightful feedback.

## Impact Statement

This paper presents work whose goal is to advance the field of Machine Learning. There are many potential societal consequences of our work, none of which we feel must be specifically highlighted here.

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

## A. The Connection between Interaction Loss and PID

We link our perturbation-based losses to components in Partial Information Decomposition (PID), following Bertschinger et al. (2014):

$$I(T; X_1, X_2) = \text{Red}(T; X_1, X_2) + \text{Unq}(T; X_1 \setminus X_2) + \text{Unq}(T; X_2 \setminus X_1) + \text{Syn}(T; X_1, X_2) \tag{4}$$

In the two-modality scenario, our model learns four experts, each trained to specialize in a PID component using corrupted modality inputs.

**Unique Information.** Experts $F_{\text{uni1}}$ and $F_{\text{uni2}}$ are trained on inputs where the other modality is replaced with noise:

$$\mathcal{L}_{\text{uni1}} = \|F_{\text{uni1}}(X_1, \tilde{X}_2) - T\|, \quad \mathcal{L}_{\text{uni2}} = \|F_{\text{uni2}}(\tilde{X}_1, X_2) - T\| \tag{5}$$

Assuming $\tilde{X}_i$ contains no task-relevant information, these losses approximate:

$$\mathcal{L}_{\text{uni1}} \propto \text{Unq}(T; X_1 \setminus X_2), \quad \mathcal{L}_{\text{uni2}} \propto \text{Unq}(T; X_2 \setminus X_1) \tag{6}$$

This aligns with unique information as defined by conditional information under fixed marginals (Bertschinger et al., 2014; Wollstadt et al., 2023).

**Redundant Information.** Expert $F_{\text{red}}$ is trained to match predictions from either single-modality input:

$$\mathcal{L}_{\text{red}} = \frac{1}{2} \left( \|F_{\text{red}}(X_1, \tilde{X}_2) - T\| + \|F_{\text{red}}(\tilde{X}_1, X_2) - T\| \right) \tag{7}$$

This loss encourages $F_{\text{red}}$ to extract information shared by both $X_1$ and $X_2$, approximating:

$$\mathcal{L}_{\text{red}} \propto \text{Red}(T; X_1, X_2) \tag{8}$$

It aligns with redundancy defined via shared informativeness (Williams & Beer, 2010; Wollstadt et al., 2023).

**Synergistic Information.** Expert $F_{\text{syn}}$ is trained to rely on both modalities jointly. It is penalized for performing well on any partial view:

$$\mathcal{L}_{\text{syn}} = \frac{1}{2} \left( \|F_{\text{syn}}(X_1, X_2) - T\| - \|F_{\text{syn}}(\tilde{X}_1, X_2) - T\| - \|F_{\text{syn}}(X_1, \tilde{X}_2) - T\| \right) \tag{9}$$

This loss isolates information that emerges only through joint modality interaction:

$$\mathcal{L}_{\text{syn}} \propto \text{Syn}(T; X_1, X_2) \tag{10}$$

This formulation reflects the formal synergy component as defined in Williams & Beer (2010); Wibral et al. (2017).

By explicitly constructing perturbed input views that suppress or preserve specific modality contributions, each expert is trained to model a distinct PID component. This forms a contrastive approximation to the constrained information projections discussed in prior work (Bertschinger et al., 2014; Williams & Beer, 2010).

## B. Empirical Evidence for the Random Vector Masking

The use of random vector replacement for modality dropout may appear ad hoc. However, our design is motivated by the need to fully suppress information from the dropped modality during interaction supervision. In contrast, alternatives such as mean or zero vector replacement risk preserving residual signals, which can undermine disentanglement of unique and redundant information pathways.

This decision is further supported by findings from CoMM (Dufumier et al., 2024), which highlight the regularization benefits and improved robustness of full modality dropout.

To assess this empirically, we conducted an ablation comparing three masking strategies—random, mean, and zero vector replacements—across five datasets. The results (Table 5) show that random vector masking consistently yields stronger performance on most metrics and tasks.

*Table 5.* Performance comparison across different modality masking strategies (Random, Mean, Zero). Metrics: Accuracy (Acc), AUROC, Micro/Macro F1. Numbers are reported as mean ± standard deviation.

| Dataset | ADNI | | MIMIC | | IMDB | | MOSI | ENRICO |
|---|---|---|---|---|---|---|---|---|
| Metric | Acc (3) | AUROC | Acc (2) | AUROC | Micro F1 (23) | Macro F1 (23) | Acc (2) | Acc (20) |
| Random | **65.08 ± 1.52** | **81.09 ± 0.02** | 69.78 ± 0.91 | **68.81 ± 0.99** | **61.00 ± 0.44** | **52.38 ± 0.48** | **71.91 ± 2.20** | 48.22 ± 1.61 |
| Mean | 59.85 ± 3.52 | 76.40 ± 2.84 | **70.00 ± 1.27** | 67.96 ± 1.43 | 59.36 ± 0.14 | 50.82 ± 0.46 | 68.95 ± 2.37 | **50.00 ± 1.94** |
| Zero | 59.48 ± 1.61 | 77.06 ± 0.60 | 69.80 ± 0.97 | 64.62 ± 1.39 | 60.57 ± 0.07 | 51.16 ± 0.76 | 70.41 ± 0.66 | 48.63 ± 1.28 |

These results support our use of random vector masking as a more effective strategy for isolating and supervising interaction-specific information flow in multimodal learning.

## C. Complete Training Objective

Let $\{F_i\}_{i=1}^{B}$ denote the $B = n + 2$ interaction experts: $n$ uniqueness experts, one synergy expert, and one redundancy expert. For each expert $F_i$, we obtain outputs from $(1 + n)$ forward passes (one full input and one for each modality replaced):

$$[\hat{y}_i^{(0)}, \hat{y}_i^{(1)}, \ldots, \hat{y}_i^{(n)}] = F_i.\texttt{forward\_multiple}(X_1, \ldots, X_n)$$

The main prediction is computed as:

$$\hat{y} = \sum_{i=1}^{B} w_i \cdot \hat{y}_i^{(0)}, \quad \text{where } [w_1, \ldots, w_B] = \texttt{MLPReWeight}(X_1, \ldots, X_n)$$

The task loss is defined as:

$$\mathcal{L}_{\text{task}} = \ell(\hat{y}, T)$$

We define the expert-specific interaction losses as follows:

**Uniqueness loss** for each $F_i$ ($i = 1, \ldots, n$):

$$\mathcal{L}_{\text{int}}^{(i)} = \frac{1}{n-1} \sum_{j \neq i} \texttt{TripletLoss}\left(\hat{y}_i^{(0)},\ \hat{y}_i^{(j)},\ \hat{y}_i^{(i)}\right)$$

**Synergy loss** ($F_{n+1}$):

$$\mathcal{L}_{\text{int}}^{(n+1)} = \frac{1}{n} \sum_{j=1}^{n} \texttt{CosSim}\left(\texttt{normalize}(\hat{y}_{n+1}^{(0)}),\ \texttt{normalize}(\hat{y}_{n+1}^{(j)})\right)$$

**Redundancy loss** ($F_{n+2}$):

$$\mathcal{L}_{\text{int}}^{(n+2)} = \frac{1}{n} \sum_{j=1}^{n} \left(1 - \texttt{CosSim}\left(\texttt{normalize}(\hat{y}_{n+2}^{(0)}),\ \texttt{normalize}(\hat{y}_{n+2}^{(j)})\right)\right)$$

We then average the interaction loss over all experts:

$$\mathcal{L}_{\text{int}} = \frac{1}{B} \sum_{i=1}^{B} \mathcal{L}_{\text{int}}^{(i)}$$

The final training objective is:

$$\mathcal{L}_{\text{total}} = \mathcal{L}_{\text{task}} + \lambda_{\text{int}} \cdot \mathcal{L}_{\text{int}}$$

Model parameters are updated to minimize $\mathcal{L}_{\text{total}}$.

## D. Computational Overhead and Scalability

In theory, $\text{I}^2\text{MoE}$ scales linearly with the number of input modalities. Specifically, the fusion overhead increases by approximately (Numer of modalities $+2$) times, corresponding to one uniqueness expert per modality, plus one redundancy and one synergy expert.

To quantify the overhead of our method, we compare $\text{I}^2\text{MoE-MulT}$ with the MulT baseline across three key metrics: training time per epoch (in seconds), inference latency (in seconds), and parameter count. As shown in Table 6, $\text{I}^2\text{MoE}$ introduces moderate increases in compute—roughly proportional to the number of modalities plus two (accounting for synergy and redundancy experts). All experiments were run on a single NVIDIA A100 GPU. Despite this additional cost, the model yields consistent improvements in interpretability and predictive performance, justifying the added overhead.

*Table 6.* Comparison of MulT and $\text{I}^2\text{MoE-MulT}$ on training time, inference latency, and model size across datasets.

| Dataset | Modalities | Train / epoch (s) | | Inference (s) | | # Params | |
|---|---|---|---|---|---|---|---|
| | | MulT | $\text{I}^2\text{MoE-MulT}$ | MulT | $\text{I}^2\text{MoE-MulT}$ | MulT | $\text{I}^2\text{MoE-MulT}$ |
| ADNI | I, G, C, B | 8.98 ± 0.04 | 16.82 ± 0.02 | 1.34 ± 0.00 | 2.29 ± 0.00 | 1,072,131 | 6,696,728 |
| MIMIC | L, N, C | 2.24 ± 0.01 | 33.67 ± 0.67 | 0.15 ± 0.00 | 0.91 ± 0.00 | 268,034 | 1,390,095 |
| IMDB | L, I | 3.62 ± 0.00 | 44.20 ± 0.59 | 0.53 ± 0.00 | 3.23 ± 0.00 | 1,068,567 | 4,423,008 |
| MOSI | V, A, T | 0.70 ± 0.00 | 4.47 ± 0.01 | 0.09 ± 0.00 | 0.48 ± 0.00 | 134,402 | 673,935 |
| ENRICO | S, W | 1.38 ± 0.02 | 6.17 ± 0.03 | 0.20 ± 0.00 | 0.44 ± 0.00 | 538,644 | 2,352,724 |

## E. Details for Dataset Preprocessing

We followed the same preprocessing procedure of the ADNI dataset and MIMIC dataset, as described in Flex-MoE (Yun et al., 2024).

### E.1. Detailed Data Preprocessing in ADNI

**Imaging, Genetic, Biospecimen, Clinical Modalities.** The Alzheimer's Disease Initiative (ADNI) is a longitudinal multi-center observational study containing multi-modal data from subjects diagnosed as cognitively normal (CN), mild cognitive impairment (MCI), and Alzheimer's dementia (AD) (Weiner et al., 2010; 2017). In our experiments, we utilized imaging, genetic, biospecimen, and clinical modalities. The imaging data consisted of magnetic resonance images (MRIs) which were preprocessed using field intensity inhomogeneity correction, gray tissue matter segmentation via MUSE (Multiatlas Region Segmentation Utilizing Ensembles of Registration Algorithms and Parameters) (Doshi et al., 2016), and voxel-wise volumetric mapping of tissue regions. The genetic data consisted of SNP (single nucleotide polymorphisms) data from the ADNI 1, GO/2, and 3 studies. These were preprocessed via alignment to a unified reference, followed by aligning strands based on the 1000 Genome Project phase 3, linkage disequilibrium (LD) pruning, and imputation. The resulting data consisted of $144,746$ SNPs. The biospecimen modality included CSF A$\beta$1-42 and A$\beta$1-40, Total Tau and Phosphorylated Tau, Plasma Neurofilament Light Chain, and ApoE genotype. Clinical data included medical history, neurological exams, patient demographics, medications, and vital signs. Data columns directly containing Alzheimer's Disease diagnosis information were excluded. For both biospecimen and clinical data, numerical data was scaled using a MinMax scaler to a range of -1 to 1, while categorical data was one-hot encoded. Missing values, were imputed using the mean for numerical fields and the mode for categorical fields.

### E.2. Detailed Data Preprocessing in MIMIC

**Lab, Notes, Codes Modalities.** The MIMIC dataset was extracted from the Medical Information Mart for Intensive Care IV (MIMIC-IV) database, which contains de-identified health data for patients who were admitted to either the emergency

department or stayed in critical care units of the Beth Israel Deaconess Medical Center in Boston, Massachusetts (Johnson et al., 2024; 2023; Goldberger et al., 2000). MIMIC-IV excludes patients under 18 years of age. We take a subset of the MIMIC-IV data, where each patient has at least more than 1 visit in the dataset as this subset corresponds to patients who likely have more serious health conditions. For each datapoint, we extract ICD-9 codes, clinical text, and labs and vital values. Using this data, we perform binary classification on one-year mortality. We drop visits that occur at the same time as the patient's death.

## F. Details for Modality-specific Encoder and Classification Head

❶ **ADNI Dataset**: For the image modality, we employed a customized 3D-CNN (Esmaeilzadeh et al., 2018) with a hidden dimension of 256 as the encoder. For the genomics, clinical, and biospecimen modalities, we used a one-hidden-layer MLP with a hidden dimension of 256 as the encoder.

❷ **MIMIC Dataset**: For all lab, note, and code modalities, we utilized an LSTM with a hidden dimension of 256 as the encoder.

❸ **MOSI Dataset**: A Gated Recurrent Unit (GRU) with a hidden dimension of 256 was used as the encoder for the vision, audio, and text modalities.

❹ **ENRICO Dataset**: For both the screenshot image and wireframe image modalities, we used VGG11 from the torchvision library with a hidden dimension size of 16 as the encoder.

❺ **IMDB Dataset**: For the image modality, a VGG-16 model was applied as the feature extractor. For the language modality, features were extracted using the pretrained Google Word2vec model. Additionally, we employed VGG11 from the torchvision library with a hidden dimension size of 16 as the encoder and used MaxoutLinear unimodal encoders, following current work (Liang et al., 2021).

▷ **Classification Head**: For all models and all datasets, we use a linear classification head to output the corresponding prediction.

## G. Details for Hyperparameter Setting

To improve reproducibility, the tables below provide a summary of the hyperparameters used in our experiments. For hyperparameters of other baseline fusion methods, please refer to the scripts in the GitHub repository at `https://github.com/Raina-Xin/I2MoE/tree/main/scripts/train_scripts`.

*Table 7.* Hyperparameter Configuration for `I²MoE-MulT` on Different Datasets

| Hyperparameter | ADNI | MIMIC | IMDB | MOSI | ENRICO |
|---|---|---|---|---|---|
| Learning Rate (`lr`) | 0.0001 | 0.0001 | 0.0001 | 0.0001 | 0.0001 |
| Temperature for Reweighting (`temperature_rw`) | 1 | 2 | 2.0 | 2.0 | 2.0 |
| Hidden Dimension for Reweighting (`hidden_dim_rw`) | 256 | 128 | 256 | 256 | 256 |
| Number of Layers in Reweighting (`num_layer_rw`) | 2 | 2 | 3 | 3 | 3 |
| Interaction Loss Weight (`interaction_loss_weight`) | 0.5 | 0.01 | 0.5 | 0.005 | 0.5 |
| Modality (`modality`) | IGCB | LNC | LI | TVA | SW |
| Training Epochs (`train_epochs`) | 50 | 30 | 40 | 30 | 50 |
| Batch Size (`batch_size`) | 32 | 32 | 32 | 32 | 32 |
| Number of Experts (`num_experts`) | 8 | 4 | 4 | 4 | 4 |
| Number of Layers in Encoder (`num_layers_enc`) | 1 | 1 | 1 | 1 | 2 |
| Number of Layers in Fusion (`num_layers_fus`) | 2 | 2 | 2 | 1 | 2 |
| Number of Layers in Prediction (`num_layers_pred`) | 2 | 2 | 2 | 1 | 2 |
| Number of Attention Heads (`num_heads`) | 4 | 1 | 4 | 1 | 4 |
| Hidden Dimension (`hidden_dim`) | 256 | 128 | 256 | 256 | 256 |
| Number of Patches (`num_patches`) | 16 | 8 | 4 | 4 | 8 |

*Table 8.* Hyperparameter Configuration for `I²MoE-SwitchGate` on Different Datasets

| Hyperparameter | ADNI | MIMIC | IMDB | MOSI | ENRICO |
|---|---|---|---|---|---|
| Learning Rate (`lr`) | 0.0001 | 0.0001 | 0.0001 | 0.0001 | 0.0001 |
| Temperature for Reweighting (`temperature_rw`) | 2 | 2 | 2.0 | 2.0 | 1 |
| Hidden Dimension for Reweighting (`hidden_dim_rw`) | 256 | 256 | 256 | 128 | 128 |
| Number of Layers in Reweighting (`num_layer_rw`) | 2 | 2 | 2 | 1 | 3 |
| Interaction Loss Weight (`interaction_loss_weight`) | 0.01 | 0.5 | 0.5 | 0.001 | 0.01 |
| Modality (`modality`) | IGCB | LNC | LI | TVA | SW |
| Training Epochs (`train_epochs`) | 30 | 30 | 40 | 50 | 30 |
| Batch Size (`batch_size`) | 8 | 64 | 64 | 32 | 8 |
| Number of Experts (`num_experts`) | 16 | 16 | 16 | 4 | 4 |
| Number of Layers in Encoder (`num_layers_enc`) | 2 | 2 | 2 | 1 | 1 |
| Number of Layers in Fusion (`num_layers_fus`) | 2 | 2 | 2 | 1 | 1 |
| Number of Layers in Prediction (`num_layers_pred`) | 2 | 2 | 2 | 1 | 1 |
| Number of Attention Heads (`num_heads`) | 4 | 4 | 4 | 4 | 2 |
| Hidden Dimension (`hidden_dim`) | 128 | 256 | 128 | 128 | 128 |
| Number of Patches (`num_patches`) | 8 | 16 | 4 | 16 | 4 |

*Table 9.* Hyperparameter Configuration for `I²MoE-InterpretCC` on Different Datasets

| Hyperparameter | ADNI | MIMIC | IMDB | MOSI | ENRICO |
|---|---|---|---|---|---|
| Learning Rate (`lr`) | 0.0001 | 0.0001 | 0.0001 | 0.0001 | 0.0001 |
| Temperature for Reweighting (`temperature_rw`) | 2 | 2 | 2.0 | 1.5 | 4.0 |
| Hidden Dimension for Reweighting (`hidden_dim_rw`) | 128 | 128 | 256 | 256 | 256 |
| Number of Layers in Reweighting (`num_layer_rw`) | 2 | 2 | 3 | 2 | 2 |
| Interaction Loss Weight (`interaction_loss_weight`) | 0.5 | 0.1 | 0.01 | 0.001 | 0.5 |
| Modality (`modality`) | IGCB | LNC | LI | TVA | SW |
| Tau ($\tau$) | 1.0 | 0.7 | 1.0 | 1.0 | 0.5 |
| Threshold (`threshold`) | 0.5 | 0.5 | 0.6 | 0.6 | 0.4 |
| Train Epochs (`train_epochs`) | 30 | 50 | 40 | 50 | 60 |
| Batch Size (`batch_size`) | 32 | 128 | 32 | 32 | 64 |
| Hidden Dimension (`hidden_dim`) | 128 | 256 | 256 | 128 | 256 |
| Hard (`hard`) | True | True | True | True | True |

*Table 10.* Hyperparameter Configuration for `I²MoE-MoE++` on Different Datasets

| Hyperparameter | ADNI | MIMIC | IMDB | MOSI | ENRICO |
|---|---|---|---|---|---|
| Learning Rate (`lr`) | 0.0001 | 0.0001 | 0.0001 | 0.0001 | 0.0001 |
| Temperature for Reweighting (`temperature_rw`) | 2 | 1 | 1.0 | 2 | 1 |
| Hidden Dimension for Reweighting (`hidden_dim_rw`) | 256 | 256 | 256 | 128 | 256 |
| Number of Layers in Reweighting (`num_layer_rw`) | 3 | 2 | 2 | 2 | 2 |
| Interaction Loss Weight (`interaction_loss_weight`) | 0.5 | 0.5 | 0.5 | 0.001 | 0.5 |
| Modality (`modality`) | IGCB | LNC | LI | TVA | SW |
| Training Epochs (`train_epochs`) | 50 | 30 | 40 | 50 | 50 |
| Batch Size (`batch_size`) | 64 | 32 | 32 | 32 | 32 |
| Number of Experts (`num_experts`) | 8 | 4 | 4 | 8 | 8 |
| Number of Layers in Encoder (`num_layers_enc`) | 2 | 2 | 2 | 2 | 2 |
| Number of Layers in Fusion (`num_layers_fus`) | 2 | 2 | 2 | 1 | 2 |
| Number of Layers in Prediction (`num_layers_pred`) | 2 | 2 | 2 | 2 | 2 |
| Number of Attention Heads (`num_heads`) | 4 | 4 | 4 | 4 | 4 |
| Hidden Dimension (`hidden_dim`) | 256 | 128 | 256 | 64 | 64 |
| Number of Patches (`num_patches`) | 8 | 4 | 8 | 4 | 4 |

## H. Human Evaluation for Local Interpretation

To strengthen evidence for the local interpretability of our model, we conducted a human evaluation study involving 15 participants. Each participant was shown 20 movie examples, resulting in a total of 300 interaction expert weight evaluations. Participants were asked to assess how reasonable the model's assigned expert weights were, choosing from a 5-point Likert scale: "Completely makes sense," "Mostly makes sense," "Neutral," "Makes little sense," and "Makes no sense at all."

Overall, 70.4% of responses were positive (i.e., "Mostly makes sense" or "Completely makes sense"), while only 9% were negative. Notably, just 0.7% of ratings selected the lowest option. These results suggest that the model's expert weight assignments are broadly viewed as reasonable and interpretable by human evaluators.

The questionnaire and de-identified responses are available at `https://github.com/Raina-Xin/I2MoE/tree/main/assets/human_eval`

*Table 11.* Distribution of human ratings for local interaction expert weights ($n = 300$).

| Response Option | Percentage of Responses |
|---|---|
| Completely makes sense | 19.4% |
| Mostly makes sense | 51.0% |
| Neutral | 19.7% |
| Makes little sense | 9.0% |
| Makes no sense at all | 0.7% |

# I. More Qualitative Examples for Local Interpretation

We present a comprehensive visualization of all 23 classes in the IMDB dataset, illustrating local interpretability for individual examples. All examples are correctly predicted by I²MoE.

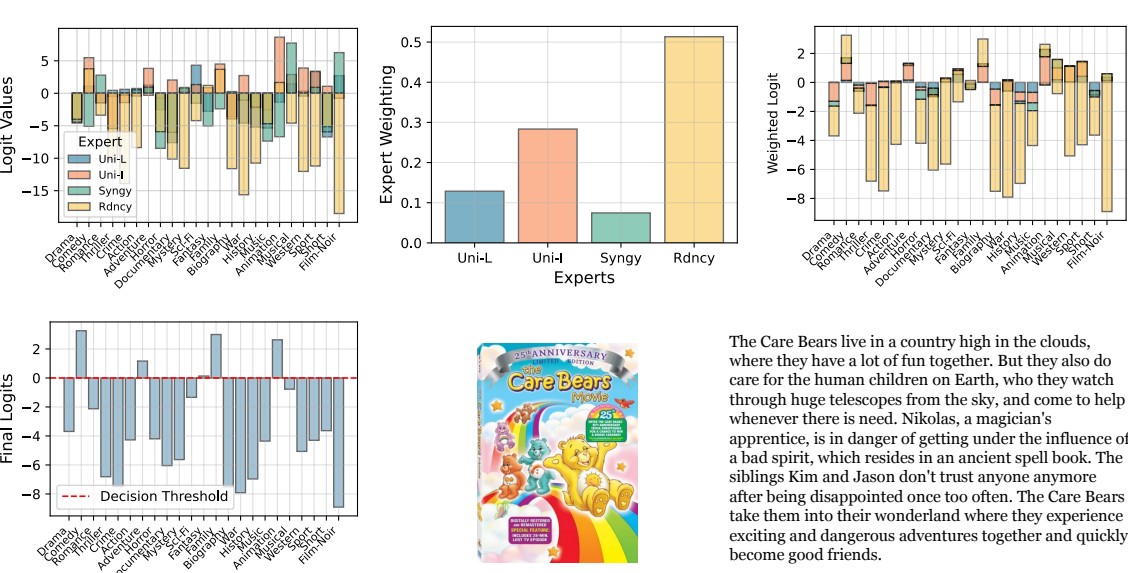

*Figure 6.* IMDB example (ID: 0088885).

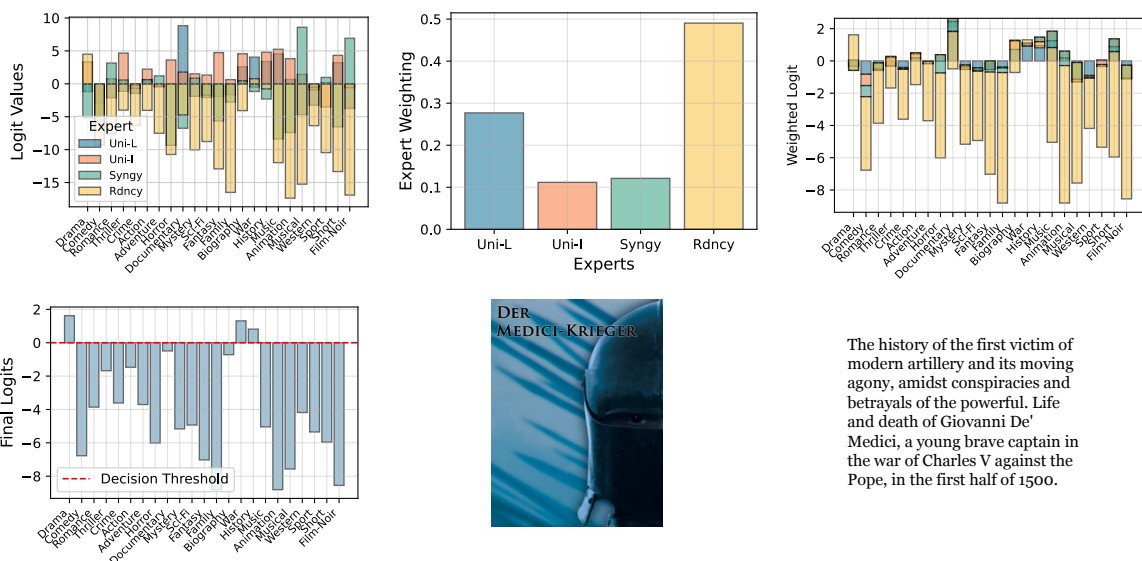

The history of the first victim of modern artillery and its moving agony, amidst conspiracies and betrayals of the powerful. Life and death of Giovanni De' Medici, a young brave captain in the war of Charles V against the Pope, in the first half of 1500.

*Figure 7.* IMDB example (ID: 0245276).

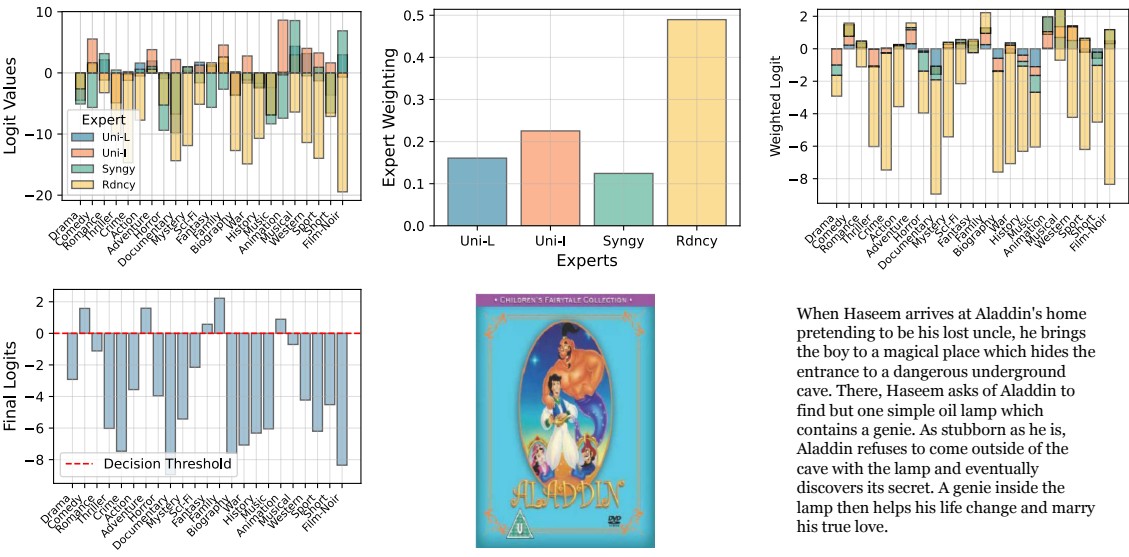

When Haseem arrives at Aladdin's home pretending to be his lost uncle, he brings the boy to a magical place which hides the entrance to a dangerous underground cave. There, Haseem asks of Aladdin to find but one simple oil lamp which contains a genie. As stubborn as he is, Aladdin refuses to come outside of the cave with the lamp and eventually discovers its secret. A genie inside the lamp then helps his life change and marry his true love.

*Figure 8.* IMDB example (ID: 0827990).

