# OpenReview forum: "$\texttt{I$^2$MoE}$: Interpretable Multimodal Interaction-aware Mixture-of-Experts"
_ICML.cc/2025/Conference — ICML 2025 poster_

### Official Review · Reviewer_aVYo · 2025-03-10

**Overall Recommendation:** 3

**Summary:**

The paper introduces a new mixture-of-experts framework designed to explicitly model diverse modality interactions and multi-modality fusion, through specialized parameters and weakly-supervised interaction losses. The proposed method is validated on five multimodal datasets, across different modalities, showing state-of-the-art performances.

**Claims And Evidence:**

1. For using triplet margin loss to model uniqueness interactions, what is the margin used in the work, why is the margin loss used here?
2. The author should discuss the computation overhead for the proposed method, compared with its counterparts, such as I2MoE-MulT vs. MulT.
3. MMoE is a very close work to the proposed method, but it is not well compared in the experiment, are there any special concerns about not comparing with it?
4. In section 6.2, how to determine the agreement/disagreement between experts, do the authors use any thresholds?

**Essential References Not Discussed:**

N/A

**Experimental Designs Or Analyses:**

Yes, the experimental design and analysis are sound

**Methods And Evaluation Criteria:**

The proposed methods and evaluation look sound

**Other Comments Or Suggestions:**

N/A

**Other Strengths And Weaknesses:**

N/A

**Questions For Authors:**

N/A

**Relation To Broader Scientific Literature:**

It is a very interesting topic to achive interpretable interaction/fusion on multi-modality data.

**Theoretical Claims:**

There are no issues regarding the theoretical claims

---

> ### Author Rebuttal · Authors · 2025-04-01
>
> Thanks for your encouraging feedback. Point-to-point responses below.
>
> >Q1. For using triplet margin loss to model uniqueness interactions, what is the margin used in the work, why is the margin loss used here?
>
> We use triplet margin loss to uniqueness interactions, as it naturally aligns with the modeling of uniqueness—there are both positive and negative examples. For example, for uniqueness expert for modality 1, we treat the full input as the anchor, the one input with modality 1 being masked as a negative, and all other perturbed inputs as positives.
>
> Triplet margin loss maximizes the distance between the anchor and a negative and minimizes the distance between an anchor and a positive. We set the margin to 1.0, selected based on validation performance. A sweep across three datasets showed that margin = 1.0 performs best in nearly all cases (see table below).
>
> | Dataset | ADNI        |        | MIMIC       |        | ENRICO        |
> |--------|-------------|--------|-------------|--------|---------------|
> | **Margin** | **Acc (3)**     | **AUROC**  | **Acc (2)**     | **AUROC**  | **Acc (20)**  |
> | 0.2 | 64.05 ± 1.97 | 80.47 ± 0.95 | 68.07 ± 2.07 | 69.42 ± 1.65 | 46.58 ± 1.75 |
> | 0.5 | 63.49 ± 1.08 | 81.05 ± 0.83 | 69.21 ± 0.61 | 68.53 ± 0.65 | 47.72 ± 1.78 |
> | **1 (Current)** | **65.08 ± 1.52** | **81.09 ± 0.02** | **69.78 ± 0.91** | **68.81 ± 0.99** | **48.22 ± 1.61** |
> | 2 | 62.56 ± 2.06 | 79.49 ± 1.82 | 68.76 ± 1.16 | 69.97 ± 0.42 | 46.80 ± 1.71 |
> | 5 | 63.96 ± 0.95 | 80.47 ± 1.25 | 68.12 ± 1.86 | 68.07 ± 1.34 | 47.95 ± 1.01 |
>
>
> >Q2. The author should discuss the computation overhead for the proposed method, compared with its counterparts, such as I2MoE-MulT vs. MulT
>
> While I2MoE introduces additional forward passes—scaling linearly with the number of modalities—the method remains fully end-to-end and weakly supervised. The fusion overhead increases by approximately (#modalities + 2), corresponding to the number of specialized experts.
>
> To quantify the overhead, we compare I2MoE-MulT to MulT across training time per epoch, inference latency, and parameter count (see table below). All experiments are conducted on a single A100 GPU. Despite moderate increases, we find the added cost justified by the significant gains in interpretability and predictive performance.
>
> | Dataset | ADNI | MIMIC | IMDB | MOSI | ENRICO |
> |---------|------|--------|----------|-----------|--------|
> | **Modality** | I,G,C,B | L,N,C | L,I | V,A,T | S,W |
> | **Train per epoch (s)** | | | | | |
> | MulT | 8.98 ± 0.04 | 2.24 ± 0.01 | 3.62 ± 0.00 | 0.70 ± 0.00 | 1.38 ± 0.02 |
> | I2MoE-MulT | 16.82 ± 0.02 | 33.67 ± 0.67 | 44.20 ± 0.59 | 4.47 ± 0.01 | 6.17 ± 0.03 |
> | **Inference (s)** | | | | | |
> | MulT | 1.34 ± 0.00 | 0.15 ± 0.00 | 0.53 ± 0.00 | 0.09 ± 0.00 | 0.20 ± 0.00 |
> | I2MoE-MulT | 2.29 ± 0.00 | 0.91 ± 0.00 | 3.23 ± 0.00 | 0.48 ± 0.00 | 0.44 ± 0.00 |
> | **# Parameters** | | | | | |
> | MulT | 1,072,131 | 268,034 | 1,068,567 | 134,402 | 538,644 |
> | I2MoE-MulT | 6,696,728 | 1,390,095 | 4,423,008 | 673,935 | 2,352,724 |
>
>
> >Q3. MMoE is a very close work to the proposed method, but it is not well compared in the experiment, are there any special concerns about not comparing with it?
>
> While both MMoE and I2MoE aim to model heterogeneous interactions, they differ substantially in design and applicability, making direct comparison non-trivial:
> 1. **End-to-End vs. Preprocessing Dependency**: I2MoE is fully end-to-end, with interaction specialization emerging via weak supervision. MMoE relies on a seperate preprocessing step to cluster training data by interaction type, breaking end-to-end training.
> 2. **Local Interpretability**: I2MoE provides instance-level interpretability by quantifying expert contributions per sample. MMoE lacks this capability, limiting its utility in settings requiring explanation of individual predictions.
> 3. **Generalizability to Higher Modalities and Complex Domains**: MMoE is tailored to vision-language tasks with pretrained LLM/VLM backbones. Its extension to domains like healthcare—where pretrained models for structured data or multi-way modality combinations are lacking—is unclear. In contrast, I2MoE operates without modality-specific pretraining and supports >2 modalities, as demonstrated on ADNI and MIMIC datasets.
>
> >Q4. In section 6.2, how to determine the agreement/disagreement between experts, do the authors use any thresholds?
>
> We define expert agreement based on predicted labels. For single-label classification tasks (i.e., ADNI, MIMIC, ENRICO), each expert outputs a logit vector, and predictions are obtained via argmax. For multi-label classification tasks (i.e., MM-IMDB), predictions are thresholded at 0.5 after sigmoid activation. For the regression task (i.e., CMU-MOSI), each expert outputs a real number the threshold is set at 0. Agreement occurs when all experts predict the same label(s); disagreement is defined as any mismatch between expert outputs.

---

### Official Review · Reviewer_LfMU · 2025-03-13

**Overall Recommendation:** 2

**Summary:**

This paper addresses multimodal learning using mixture-of-experts, where dedicated experts learn distinct information from input modalities. The authors introduce a reweighting model to interpretably assign weights to the experts, facilitating understanding of their individual importance. The proposed approach is evaluated on five multimodal benchmarks covering various modalities and tasks.

**Claims And Evidence:**

The related work section (lines 93-94) mentions only one previous work applying mixture-of-expert (MoE) to multimodal learning. However, some related studies might have been overlooked, such as [1][2].

[1] Wu, Mike, and Noah Goodman. "Multimodal generative models for scalable weakly-supervised learning." Advances in neural information processing systems 31 (2018).

[2] Shi, Yuge, Brooks Paige, and Philip Torr. "Variational mixture-of-experts autoencoders for multi-modal deep generative models." Advances in neural information processing systems 32 (2019).

**Essential References Not Discussed:**

Please see references noted in the "Claims and Evidence" section above.

**Experimental Designs Or Analyses:**

Overall, the experimental design and analyses appear sound, but several aspects could benefit from clarification:
+ In Table 1, the reported results on MM-IMDB (Micro F1: 61.00; Macro F1: 52.38) seem notably lower compared to previous literature such as MFAS [3] (Macro F1: 55.70), CentralNet [4] (Micro F1: 63.90; Macro F1: 56.10), and ViLT [5] (Micro F1: 64.70; Macro F1: 55.30). Given that these methods employ relatively straightforward fusion strategies (e.g., early fusion), could the authors discuss why their proposed MoE approach achieves lower scores?
+ The reported MM-IMDB result in Figure 5 (49.21) appears different from the value in Table 1 (52.38). Could the authors clarify this inconsistency?
+ In Figure 3(b), the unique information learned from language modality appears to have lower weights. Does this suggest language is less important for MM-IMDB? It would be valuable to have more insight or discussion on this observation.
+ The proposed method requires multiple runs during training, which could significantly increase computational costs. It would be helpful if the authors could report metrics related to computational efficiency, such as training time.
+ Additionally, the studied baselines such as VGG and LSTM seem somewhat outdated. Incorporating more recent baseline methods would strengthen the evaluation.

[3] Pérez-Rúa, Juan-Manuel, et al. "MFAS: Multimodal fusion architecture search." Proceedings of the IEEE/CVF Conference on Computer Vision and Pattern Recognition. 2019.

[4] Vielzeuf, Valentin, et al. "Centralnet: a multilayer approach for multimodal fusion." Proceedings of the European Conference on Computer Vision (ECCV) Workshops. 2018.

[5] Ma, Mengmeng, et al. "Are multimodal transformers robust to missing modality?." Proceedings of the IEEE/CVF conference on computer vision and pattern recognition. 2022.

**Methods And Evaluation Criteria:**

Could the authors clarify their motivation for using random vectors to mask modalities? There is concern that using random vectors might add noise to the fused representations. Would modality mean vectors potentially offer a less noisy alternative? Additional clarification on your design choice here would strengthen the paper.

**Other Comments Or Suggestions:**

Some dataset names appear inconsistent (e.g., "MM-IMDB" vs. "MMIMDB").

**Other Strengths And Weaknesses:**

Strengths:
+ The paper is generally well-organized and clear.
+ The problem of interpretable multimodal fusion is meaningful and relevant to a broad audience.

Weaknesses:
- Please refer to the comments provided above for potential improvements.

**Questions For Authors:**

The following clarifications would help strengthen the manuscript:
+ Could you provide insights on why your approach yields lower performance compared to simpler fusion methods in existing literature?
+ Can you explain the inconsistency between the MM-IMDB results reported in Table 1 and Figure 5?
+ Please further elaborate on your insight of using random vector masking for modalities. What are the potential limitations or implications of this design choice?
+ For more questions, see the above sections.

**Relation To Broader Scientific Literature:**

The idea of interpretable multimodal fusion is valuable, especially in safety-critical domains such as healthcare.

**Theoretical Claims:**

All provided equations appear correct.

---

> ### Author Rebuttal · Authors · 2025-04-01
>
> Thanks for your thoughtful feedback. Point-to-point responses below. **All supplementary on [GitHub](https://anonymous.4open.science/r/I2MoE-rebuttal-8308/README.md)**.
>
> >Q1. I2MoE lower scores on MM-IMDB?
>
> This is primarily due to **differences in experimental setups**:
> 1. Evaluation Setup: Our experiments follow the setup in MultiBench [1]. Within this framework, I2MoE achieves Micro-F1: 61.00 and Macro-F1: 52.38, outperforming the SOTA (Micro: 59.3, Macro: 50.2; see Table 23 in [1]).
> 2. Reimplement: While [2–4] report higher MM-IMDB scores, they adopt different experimental setups and do not release code for exact reproduction. We reimplemented CentralNet and ViLT under the MultiBench setting and observed performance drops compared to their original reports.
> 3. I2MoE Improvement: We further evaluated I2MoE as a drop-in framework applied to CentralNet and MulT. As shown below, I2MoE consistently improves performance across both Micro and Macro F1 under the MultiBench setup.
>
> | Model | **Micro F1**  | **Macro F1** |
> |--------------------|------------------|------------------|
> | CentralNet | 58.57 ± 0.58 | 49.90 ± 0.24  |
> | I2MoE-CentralNet | 58.72 ± 0.13  | 50.13 ± 0.21 |
> | ViLT  | 58.38 ± 0.44 | 48.31 ± 0.32 |
> | I2MoE-ViLT  | 59.53 ± 1.84     | 49.66 ± 2.23 |
> | MulT  | 59.68 ± 0.19     | 51.41 ± 0.04 |
> | **I2MoE-MulT**   | **61.00 ± 0.44** | **52.38 ± 0.48** |
>
> >Q2. MM-IMDB result in Figure 5 appears different from Table 1
>
> Thanks for catching this–we forgot to update the MM-IMDB result in Figure 5. Updated figures can be found on GitHub.
>
> >Q3. …motivation for using random vectors to mask modalities? ..modality mean vectors..?
>
> We use random vectors to completely remove information from the masked modality so that any observed uniqueness and synergistic interaction from the masked modality cannot be attributed to leakage.
>
> While mean vectors could be less noisy, they are also dynamic in our setting. Since I2MoE is trained end-to-end with modality-specific encoders, the mean vectors evolve across training epochs. To mitigate this instability, we maintain a running average of mean vectors (see GitHub), but this introduces additional complexity and still does not guarantee full information removal.
>
> Further, we compare random / mean / zero vectors across five datasets. As shown below, random vectors consistently outperform mean vectors in nearly all settings.
>
> | | ADNI | | MIMIC | | IMDB | | MOSI | ENRICO |
> |--------------|---------------|--------------|---------------|--------------|----------------|--------------|---------------|----------------|
> | | **Acc** | **AUROC** | **Acc** | **AUROC** | **Micro F1** | **Macro F1** | **Acc** | **Acc** |
> | **Random** | **65.08 ± 1.52** | **81.09 ± 0.02** | 69.78 ± 0.91 | **68.81 ± 0.99** | **61.00 ± 0.44**  | **52.38 ± 0.48** | **71.91 ± 2.20**  | 48.22 ± 1.61   |
> | **Mean** | 59.85 ± 3.52  | 76.40 ± 2.84 | **70.00 ± 1.27** | 67.96 ± 1.43 | 59.36 ± 0.14  | 50.82 ± 0.46 | 68.95 ± 2.37  | **50.00 ± 1.94** |
> | **Zero** | 59.48 ± 1.61 | 77.06 ± 0.60 | 69.80 ± 0.97 | 64.62 ± 1.39 | 60.57 ± 0.07  | 51.16 ± 0.76 | 70.41 ± 0.66  | 48.63 ± 1.28   |
>
> >Q4. Interpretation of Figure 3b
>
> Figure 3b shows a local (sample-level) decomposition for a specific MM-IMDB example, where the language modality contributes less unique information for that instance. This does not reflect global trends in the MM-IMDB dataset. For dataset-level insights, we refer the reviewer to Figure 4, which summarizes expert weights across the entire test set.
>
> >Q5. (I2MoE) could significantly increase computational costs
>
> Please see our response to Reviewer aVYo Q2.
>
> >Q6. The studied baselines such as VGG and LSTM seem somewhat outdated
>
> We clarify that VGG and LSTM are used only as modality-specific encoders, not as fusion architectures. This setup follows widely adopted practice in multimodal learning benchmarks such as [1], where lightweight encoders are used to ensure a fair comparison of fusion methods. We agree that powerful modality-specific encoders could further boost performance. Due to time constraints, we plan to incorporate them in the final version.
>
> >C1. Some related studies might have been overlooked
>
> We note that the two works mentioned focus on generative modeling, but we focus on interpretable discriminative modeling. We consider these lines of research orthogonal but will clarify this distinction and cite these works in our related work section.
>
> >C2. Some dataset names appear inconsistent (e.g., "MM-IMDB" vs. "MMIMDB").
>
> We will standardize all occurrences to “IMDB” in the final version.
>
> [1] Liang et al., 2021. *MultiBench: Multiscale Benchmarks for Multimodal Representation Learning*. NeurIPS.
> [2] Pérez-Rúa et al., 2019. *MFAS: Multimodal Fusion Architecture Search*. CVPR.
> [3] Vielzeuf et al., 2018. *CentralNet: A Multilayer Approach for Multimodal Fusion*. ECCV Workshops.
> [4] Ma et al., 2022. *Are Multimodal Transformers Robust to Missing Modality?*. CVPR.

---

### Official Review · Reviewer_zoTN · 2025-03-14

**Overall Recommendation:** 4

**Summary:**

In this paper, the authors introduced I2MOE, a novel multimodal model that trains a different set of experts to model each type of multimodal interaction between modalities. Each expert is trained with a different interaction loss specifically designed for the type of interaction it has to deal with, in addition to the main task objective. The model is evaluated on 5 different multimodal tasks, where the model achieves top performance on all of them. The authors also performed additional analysis or demonstrations to show that I2MOE can generalize to different fusion backbones, offers local and global interpretability, and works with more than 2 modalities. There is also ablation studies to justify the design choices of I2MOE.

**Claims And Evidence:**

The claims made in the submission is supported by clear and convincing evidence from the experiments.

The only minor problem is that, the support for "local interpretation" is only backed by qualitative samples from one task (mm-imdb) in both main text and appendix. The support for the local interpretability would be stronger with either a human evaluation of interpretability (i.e. whether the interaction attributions generated by I2MOE makes sense) or by including more qualitative samples from tasks other than mm-imdb.

**Essential References Not Discussed:**

N/A

**Experimental Designs Or Analyses:**

I have checked the soundness and validity of experimental designs, including the main experiment with 5 tasks, and additional analysis and ablations. The experimental designs looks valid and sound to me.

**Methods And Evaluation Criteria:**

The proposed methods and evaluation criteria makes sense for the problem.

**Other Comments Or Suggestions:**

N/A

**Other Strengths And Weaknesses:**

Additional Strength:

- The presentation quality of the paper is good. The methodology is easy to follow and the intuition behind each decision is clearly explained.

- There is ablation studies that clearly demonstrates the need for each design choice of the proposed method.

Additional Weakness:

- The font size of the tables and figures are tiny, making them hard to read.

- The paper did not specify the complete objective (i.e. the final combination of all task loss and interaction losses). Maybe writing down the complete objective as a mathematical expression (or maybe an algorithm block for the entire training objective) would make things more clear. For example, currently it is not clear how different losses are combined, or whether they are weighted during the combination.

**Questions For Authors:**

Can you clarify how different interaction losses are combined with the main task objectives? If they are added together, how is each loss weighted?

**Relation To Broader Scientific Literature:**

Compared to existing works in multimodal machine learning and multimodal interaction quantification/interpretability, the key contribution of this paper is that the proposed method creates one single end-to-end model that (1) achieves high performance in tasks, (2) inherently offers multimodal interaction quantification and interpretability, (3) applies to tasks with more than 2 modalities, and (4) generalizes well to different types of multimodal fusion. While previous works have proposed models or methods that can achieve some of the above, the proposed method seem to be the first that can achieve all of them within one single end-to-end model.

**Theoretical Claims:**

No theoretical claim.

---

> ### Author Rebuttal · Authors · 2025-04-01
>
> Thanks for your positive feedback. Point-to-point responses below.
>
> > Q1.The support for "local interpretation" is only backed by qualitative samples from one task
>
> Thanks for suggesting to strengthen the evidence for local interpretability. We conducted a human evaluation with 15 participants on 20 movie examples (300 total ratings), asking how reasonable the model’s assigned interaction expert weights were. Participants chose from five options, ranging from “Makes no sense at all” to “Completely makes sense.”
>
> 70.4% of responses were positive (Mostly or Completely makes sense), while only 9% were negative, and just 0.7% selected the lowest rating. These results suggest that the model’s expert weights are broadly viewed as reasonable and interpretable by human evaluators.
> | **Response**                     | **Percentage of all responses (n=300)** |
> |-----------------------------|--------------------------------------|
> | 'Completely makes sense'    | 19.7%                                |
> | 'Mostly makes sense'        | 51%                                  |
> | 'Neutral'                   | 19.7%                                |
> | 'Makes little sense'        | 9%                                   |
> | 'Makes no sense at all'     | 0.7%                                 |
>
>
> Link to the questionnaire: [Link](https://anonymous.4open.science/r/I2MoE-rebuttal-8308/AR-zoTH/Q1_human_eval/Human%20Evaluation%20of%20I2MoE%20Interpretability.pdf)
>
> Link to deidentified response: [Link](https://anonymous.4open.science/r/I2MoE-rebuttal-8308/AR-zoTH/Q1_human_eval/human_eval_deid.csv)
>
>
> >Q2. The font size of the tables and figures are tiny, making them hard to read.
>
> Thank you for the helpful suggestion. We have increased the font sizes in all figures and tables and will include them in the final version.
>
> Updated tables and figures: [Link](https://anonymous.4open.science/r/I2MoE-rebuttal-8308/AR-zoTH/Q2_font_size/table_4.png)
>
> >Q3. Writing down the complete objective as a mathematical expression (or maybe an algorithm block for the entire training objective)
>
> **Algorithm block** for I2MoE training and inference forward pass: [Link](https://anonymous.4open.science/r/I2MoE-rebuttal-8308/AR-zoTH/Q3_objective/i2moe_algorithm.png)
>
> Below we explain the complete objective in math expression:
> Let \\( \\{F_i\\}_{i=1}^{B} \\) denote the \\( B = n + 2 \\) interaction experts: \\( n \\) uniqueness experts, one synergy expert, and one redundancy expert. For each expert \\( F\_i \\), we obtain outputs from \\( (1 + n) \\) forward passes (one full input and one for each modality replaced):
>
> \\[
> [\\hat{y}\_i^{(0)}, \\hat{y}\_i^{(1)}, \\dots, \\hat{y}\_i^{(n)}] = F\_i\\mathrm{.forward\\_multiple}(X\_1, \\dots, X\_n)
> \\]
>
>
> The main prediction is computed as:
>
> \\[
> \\hat{y} = \\sum\_{i=1}^{B} w\_i \\cdot \\hat{y}\_i^{(0)}, \\quad \\text{where } [w\_1, \\dots, w\_B] = \text{MLPReWeight}(X\_1, \\dots, X\_n)
> \\]
>
> The task loss is defined as:
>
> \\[
> \\mathcal{L}\_{\\text{task}} = \\ell(\\hat{y}, T)
> \\]
>
> We define the expert-specific interaction losses as follows:
>
> **Uniqueness loss** for each \\( F\_i \\) (\\( i = 1, \\dots, n \\)):
>
> \\[
> \\mathcal{L}\_{\\text{int}}^{(i)} = \\frac{1}{n - 1} \\sum\_{j \\ne i} \\text{TripletLoss}\\left( \\hat{y}\_i^{(0)},\\; \\hat{y}\_i^{(j)},\\; \\hat{y}\_i^{(i)} \\right)
> \\]
>
> **Synergy loss** (\\( F\_{n+1} \\)):
>
> \\[
> \\mathcal{L}\_{\\text{int}}^{(n+1)} = \\frac{1}{n} \\sum\_{j=1}^{n} \\text{CosSim}\\left( \\text{normalize}(\\hat{y}\_{n+1}^{(0)}),\\; \\text{normalize}(\\hat{y}\_{n+1}^{(j)}) \\right)
> \\]
>
> **Redundancy loss** (\\( F\_{n+2} \\)):
>
> \\[
> \\mathcal{L}\_{\\text{int}}^{(n+2)} = \\frac{1}{n} \\sum\_{j=1}^{n} \\left( 1 - \\text{CosSim}\\left( \\text{normalize}(\\hat{y}\_{n+2}^{(0)}),\\; \\text{normalize}(\\hat{y}\_{n+2}^{(j)}) \\right) \\right)
> \\]
>
> We then average the interaction loss over all experts:
>
> \\[
> \\mathcal{L}\_{\\text{int}} = \\frac{1}{B} \\sum\_{i=1}^{B} \\mathcal{L}\_{\\text{int}}^{(i)}
> \\]
>
> The final training objective is:
>
> \\[
> \\mathcal{L}\_{\\text{total}} = \\mathcal{L}\_{\\text{task}} + \\lambda\_{\\text{int}} \\cdot \\mathcal{L}\_{\\text{int}}
> \\]
>
> Model parameters are updated to minimize \\( \\mathcal{L}\_{\\text{total}} \\).

---

> > ### Comment · Reviewer_zoTN · 2025-04-01
> >
> > Thanks for your response. My review remains positive.

---

> > > ### Author Response · Authors · 2025-04-03
> > >
> > > Dear Reviewer zoTN,
> > >
> > > Thank you for your response and continued positive evaluation.
> > >
> > > We're grateful for your constructive feedback, which helped us improve both the clarity and quality of our paper. We truly appreciate the time and effort you dedicated to reviewing our work.
> > >
> > > Best regards,
> > > Authors

---

### Official Review · Reviewer_KHiC · 2025-03-14

**Overall Recommendation:** 3

**Summary:**

The paper introduces $I^2MoE$, an end-to-end mixture-of-experts framework that explicitly models heterogeneous interactions between input modalities. By deploying specialized interaction experts (e.g., uniqueness, synergy, redundancy) and a reweighting module, $I^2MoE$ not only improves task performance—demonstrated by accuracy gains on datasets like ADNI, CMU-MOSI, and MM-IMDB—but also provides both local and global interpretability of multimodal fusion. The method leverages weakly supervised interaction losses based on modality perturbation to guide expert specialization.

**Claims And Evidence:**

The authors claim that:
- Modeling distinct multimodal interactions via dedicated experts improves predictive performance and interpretability.
- The reweighting mechanism provides sample-level and dataset-level insights.

Ablation studies support the necessity of each design component.

These claims are backed by extensive experiments (including ablations) and visualizations (e.g., Figure 3 and Table 4). However, the rationale behind the specific random vector perturbation strategy is more heuristic than theoretically grounded.

**Essential References Not Discussed:**

No

**Experimental Designs Or Analyses:**

The experimental setup is comprehensive:
- **Datasets:** Experiments span diverse domains (medical imaging, sentiment analysis, movie genre classification) demonstrating the model’s generality.
- **Ablation Studies:** Detailed ablations confirm the contribution of the interaction loss, reweighting module, and perturbation strategy.
- **Interpretability Analysis:** Visualizations of local expert contributions and global weight distributions validate the interpretability claims.

**Methods And Evaluation Criteria:**

The dual-objective loss (task loss plus interaction loss) and the use of a reweighting model are central to the design. Experiments across five datasets—with comparisons to both vanilla fusion and advanced baselines (e.g., SwitchGate, MoE++)—demonstrate consistent performance gains. One concern is the sensitivity of the method to hyperparameters, particularly in imbalanced datasets (e.g., MIMIC), where accuracy drops occur despite AUROC improvements.

**Other Comments Or Suggestions:**

A discussion on computational overhead and scalability with an increased number of modalities is encouraged.

**Other Strengths And Weaknesses:**

**Strengths:**
- **Novel Architecture:** Explicitly modeling heterogeneous interactions via specialized experts.
- **Interpretability:** Provides both local and global explanations, supported by qualitative and quantitative analyses.
- **Comprehensive Evaluation:** Extensive experiments and ablation studies across diverse datasets substantiate the claims.

**Weaknesses:**
- **Theoretical Grounding:** The connection to PID is mainly heuristic; more formal theoretical analysis would be beneficial.
- **Perturbation Method:** The use of random vector replacement for modality dropout is ad hoc and might affect stability.

**Questions For Authors:**

1. Could you elaborate on strategies to mitigate accuracy degradation in imbalanced datasets like MIMIC?
2. How does I2MoE scale when extending to more than two modalities or when applied to other multimodal tasks (e.g., video–audio fusion)?

**Relation To Broader Scientific Literature:**

The paper is well situated within the multimodal fusion literature. It contrasts with conventional fusion methods and recent MoE-based models. For instance:
- **MMoE (Yu et al., 2024):** Similar in spirit to I2MoE, this work also decomposes multimodal interactions into specialized experts.
- **MoMa (Jiang et al., 2024):** Introduces modality-aware expert routing for early fusion efficiency, offering insights into modality-specific parameter allocation
- **Interpretable Mixture of Experts for Structured Data (Ismail et al., 2022):** Although targeting tabular and time-series data, its inherently interpretable design may offer complementary perspectives for multimodal settings.
- **Dividing and Conquering a BlackBox (Ghosh et al., 2023):** Presents a divide-and-conquer approach for extracting interpretable models, highlighting expert specialization strategies .
- **LIMoE (Mustafa et al., 2022):** Uses contrastive learning in a MoE framework for vision–language tasks, emphasizing organic emergence of modality-specific experts .
Integrating these discussions would further contextualize I2MoE’s contributions.

**Theoretical Claims:**

The paper is motivated by the Partial Information Decomposition (PID) framework; however, the theoretical connection between the proposed weakly supervised interaction loss and formal PID measures is only loosely established.

---

> ### Author Rebuttal · Authors · 2025-04-01
>
> Thanks for your constructive feedback. Point-to-point responses below.
>
> >Q1. The connection between interaction loss and PID
>
>
> We would appreciate any insights from the reviewer on this point. Below, we attempt to connect each expert trained on the perturbed input views to a distinct PID component. This might form a contrastive approximation to the constrained information projections discussed in [1, 2]:
>
> \\[
> I(T; X_1, X_2) = \mathrm{Red}(T; X_1, X_2) + \mathrm{Unq}(T; X_1 \\setminus X_2) + \mathrm{Unq}(T; X_2 \\setminus X_1) + \mathrm{Syn}(T; X_1, X_2)
> \\]
>
> In the two-modality scenario, our model learns four experts, each trained to specialize in a PID component using perturbed modality inputs.
>
> **Unique Information [1, 3].** Experts \\( F\_{\text{uni}1} \\) and \\( F\_{\text{uni}2} \\) are trained on inputs where the other modality is replaced with noise:
>
> \\[
> \\mathcal{L}\_{\text{uni}1} = \\| F\_{\text{uni}1}(X_1, \\tilde{X}\_2) - T \\|, \quad
> \\mathcal{L}\_{\text{uni}2} = \\| F\_{\text{uni}2}(\\tilde{X}\_1, X_2) - T \\|
> \\]
>
> Assuming \\( \\tilde{X}\_i \\) contains no task-relevant information, these losses approximate:
>
> \\[
> \\mathcal{L}\_{\text{uni}1} \\propto \mathrm{Unq}(T; X_1 \\setminus X_2), \quad
> \\mathcal{L}\_{\text{uni}2} \\propto \mathrm{Unq}(T; X_2 \\setminus X_1)
> \\]
>
>
> **Redundant Information [2, 3].** Expert \\( F\_{\text{red}} \\) is trained to match predictions from either single-modality input:
>
> \\[
> \\mathcal{L}\_{\text{red}} = \\frac{1}{2} \\left( \\| F\_{\text{red}}(X_1, \\tilde{X}\_2) - T \\| + \\| F\_{\text{red}}(\\tilde{X}\_1, X_2) - T \\| \\right)
> \\]
>
> This loss encourages \\( F\_{\text{red}} \\) to extract information shared by both \\( X_1 \\) and \\( X_2 \\), approximating:
>
> \\[
> \\mathcal{L}\_{\text{red}} \\propto \mathrm{Red}(T; X_1, X_2)
> \\]
>
>
> **Synergistic Information [2, 4].** Expert \\( F\_{\text{syn}} \\) is trained to rely on both modalities jointly. It is penalized for performing well on any partial view:
>
> \\[
> \\mathcal{L}\_{\text{syn}} = \\frac{1}{2} \\left( \\| F\_{\text{syn}}(X_1, X_2) - T \\| - \\| F\_{\text{syn}}(\\tilde{X}\_1, X_2) - T \\| - \\| F\_{\text{syn}}(X_1, \\tilde{X}\_2) - T \\| \\right)
> \\]
>
> This loss isolates information that emerges only through joint modality interaction:
>
> \\[
> \\mathcal{L}\_{\text{syn}} \\propto \mathrm{Syn}(T; X_1, X_2)
> \\]
>
> >Q2. Random vector replacement for modality dropout
>
> Please see our response to Reviewer LfMU Q3.
>
> >Q3. Strategies to mitigate accuracy degradation in imbalanced datasets like MIMIC?
>
> In imbalanced settings like MIMIC, threshold-independent metrics such as AUROC provide a more reliable measure of performance. We mitigate class-imbalance by applying class weighting (0.25 for negative, 0.75 for positive) during training across all models.
>
> To clarify performance across classes, we report per-class accuracy, balanced accuracy, and AUROC below. I2MoE-MulT achieves higher balanced accuracy and AUROC than MulT, indicating improved positive class recognition without sacrificing majority class performance.
>
> | Metrics       | Acc (+) | Acc (-) | Bal. Acc | Avg. Acc | AUROC |
> |---------------|---------|---------|----------|----------|--------|
> | **MulT**       | 20.87   | **90.31**   | 55.59    | **74.64**    | 65.61  |
> | **i2MoE-MulT** | **50.49**   | 75.43   | **62.96**    | 69.80    | **69.44**  |
>
>
> >Q4. A discussion on computational overhead and scalability is encouraged
>
> Please see our response to Reviewer aVYo Q2.
>
> >Q5. How does I2MoE scale when extending to more than two modalities or when applied to other multimodal tasks (e.g., video–audio fusion)?
>
> Most real-world multimodal datasets ([5, 6]) involve fewer than four modalities. I2MoE performs well in such settings—for example, ADNI includes imaging, genetics, clinical tests, and biospecimens. We also evaluated I2MoE on video–audio fusion using CMU-MOSI (text, video, audio), where I2MoE improves the performance of four different baseline fusion methods (Table 1 and Table 2).
>
> >C1. I2MoE contrasts with conventional fusion methods and recent MoE-based models
>
> Thanks for the comment. We will expand Section 2 to position I2MoE within the broader MoE literature you mentioned more explicitly.
>
> [1] Bertschinger et al., 2014. *Quantifying Unique Information*. Entropy.
> [2] Williams and Beer, 2010. *Nonnegative Decomposition of Multivariate Information*. arXiv.
> [3] Wollstadt et al., 2023. *A Rigorous Information-Theoretic Definition of Redundancy and Relevancy in Feature Selection Based on (Partial) Information Decomposition*. JMLR.
> [4] Wibral et al., 2017. *Partial information decomposition as a unified approach to the specification of neural goal functions*. Brain and Cognition.
> [5] Liang et al., 2021. *MultiBench: Multiscale Benchmarks for Multimodal Representation Learning*. NeurIPS.
> [6] Liang et al., 2022. *High-Modality Multimodal Transformer: Quantifying Modality & Interaction Heterogeneity for High-Modality Representation Learning*. arXiv:2203.01311.

---

### Decision · Program_Chairs · 2025-05-01

**Decision:**

Accept (poster)

**Comment:**

This paper proposes a multimodal learning model that trains a different set of experts to model each type of multimodal interaction between modalities.

After the rebuttal, three reviewers ( KHiC, zoTN, and aVYo) gave positive ratings (3,4,3), recognizing the paper's strong performance and novel ideas.  Reviewer LfMU requested more detailed explanations of certain technique designs, as well as clarifications regarding the experimental design and analysis. The authors provided a thorough response, but Reviewer LfMU did not reply to the feedback.

The Area Chair (AC) agrees with most of the reviewers and recommends accepting the paper. Additionally, it is strongly suggested that the authors incorporate the content discussed in the feedback to make the paper more comprehensive.